

# A CF data model and implementation

David Hassell[1], Jonathan Gregory[1,2], Jon Blower[3], Bryan N. Lawrence[1], and Karl E. Taylor[4]

[1]National Centre for Atmospheric Science, Department of Meteorology, University of Reading, U.K.
[2]Met Office Hadley Centre, Exeter, U.K.
[3]Institute for Environmental Analytics, University of Reading, U.K.
[4]Program for Climate Model Diagnosis and Intercomparison, Lawrence Livermore National Laboratory, U.S.

*Correspondence to:* David Hassell (david.hassell@ncas.ac.uk)

**Abstract.** The CF (Climate and Forecast) metadata conventions are designed to promote the creation, processing and sharing of climate and forecasting data using Network Common Data Form (netCDF) files and libraries. The CF conventions provide a description of the physical meaning of data and of their spatial and temporal properties, but they depend on the netCDF file encoding which can currently only be fully understood and interpreted by someone familiar with the rules and relationships specified in the conventions documentation. To aid in development of CF-compliant software and to capture with a minimal set of elements all of the information contained in the CF conventions, we propose a formal data model for CF which is independent of netCDF and describes all possible CF-compliant data. Because such data will often be analysed and visualised using software based on other data models, we compare the CF data model with the ISO 19123 coverage model, the Open Geospatial Consortium CF netCDF standard and the Unidata Common Data Model. To demonstrate that the CF data model can in fact be implemented, we present cf-python, a Python software library that conforms to the model and can manipulate any CF-compliant dataset.

## 1   Introduction

Network Common Data Form (netCDF) supports a view of data as a collection of self-describing, portable objects that can be accessed through standardised software libraries. For climate scientists, as well as others, it has become a popular way to create, access, and share array-orientated scientific data (Rew and Davis, 1990; Rew et al., 2006). In this context, "self-describing" means that a file contains, for each data array, an associated description of what it represents scientifically, i.e. metadata. NetCDF was developed and is maintained at Unidata, part of the US University Corporation for Atmospheric Research (UCAR).

The CF (Climate and Forecast) metadata conventions (Eaton et al., 2011, http://cfconventions.org) are a set of rules for storing geoscientific data in netCDF files, with the aims of describing the data, enabling users to identify comparable data held in different files, and facilitating the development of software to extract, process, analyse and display the data. Initially CF was developed for gridded data from climate and forecast models of the atmosphere and ocean, but its use has subsequently extended to other geosciences, and to observations as well as numerical models. The use of CF is recommended where applicable by Unidata.



CF metadata is designed to be interpretable without reference to external tables, readable by humans, easily parsable by programs, and minimally redundant, which reduces the potential for inconsistencies. The development of CF began in 1999 and has proceeded incrementally, with new features added only when called for by common use-cases, and with consideration of how the change might impact data producers. Archival of data is a major purpose of netCDF, for which reason backwards compatibility is an important consideration. So far, no backwards-incompatible change has been made to the CF conventions,

meaning that a file written with a previous release of CF would still be compliant with the most recent version.

In general, CF tells data producers how they can provide information they think is important for understanding their data, but it mandates very little metadata. Projects which recommend or require the use of the CF conventions may of course impose additional requirements on data producers, as is done for instance by the Coupled Model Intercomparison Project (https://pcmdi.llnl.gov/mips/cmip5/CMIP5_output_metadata_requirements.pdf), which in its fifth phase (CMIP5) serves more

than 4.2 petabytes of CF-compliant netCDF (CF-netCDF) data.

In this paper we present a data model which, at the time of writing, is based on the the most recent version of the CF conventions (1.6). By a "data model" we mean an abstract interpretation of the data, that identifies the elements of the dataset and their scientific intent, and describes how they are related to one another and to the real or model world from which the data were derived. In practice if a scientist wants to create and subsequently analyse data in a CF-netCDF file, a data model

is necessary because it imposes the rules, constraints, and relationships connecting metadata to the data that are needed to imagine how the quantities included in the dataset should be combined and processed scientifically.

Up to now a comprehensive CF data model has not been explicitly proposed, so those writing software to interpret CF-compliant files have at least implicitly adopted data models that serve their needs, but perhaps without accommodating the full power of the conventions. This is not ideal because it impairs the linking of independently developed software tools when

together those tools might be needed for the analysis of CF data.

We believe that with an explicit, comprehensive data model, we can present the CF conventions in a manner that will lead to their being better understood, and that adherence to the data model will ensure the production of CF-compliant datasets, allow software developers to design CF-compliant data processing applications, and provide guidance during the development of future extensions to the CF conventions.

An explicit data model exists independent of any particular implementation, and is mainly used to understand the data structures and metadata that are useful in representing datasets and which can be relied on in building interfaces to other explicit data models. In general, the more complex a piece of software and the wider-spread its adoption, the more one needs an explicit and generally accepted data model. Otherwise, divergent interpretations of the data model will inevitably lead to misunderstandings, inconsistencies and inefficiencies.

The netCDF interface that underlies CF has an explicit data model (the yellow layer in figure 1) but currently, in the absence of an explicit data model, CF exists only in the form of netCDF conventions (the blue layer). The conventions have been widely adopted, and there are many software applications that work with CF datasets, each of them representing CF data and metadata according to some individually-assumed, implicit data model. Our aim is to create an explicit data model for CF (the green layer in figure 1) that can provide a definitive and comprehensive structural interpretation of CF. This by no means precludes a



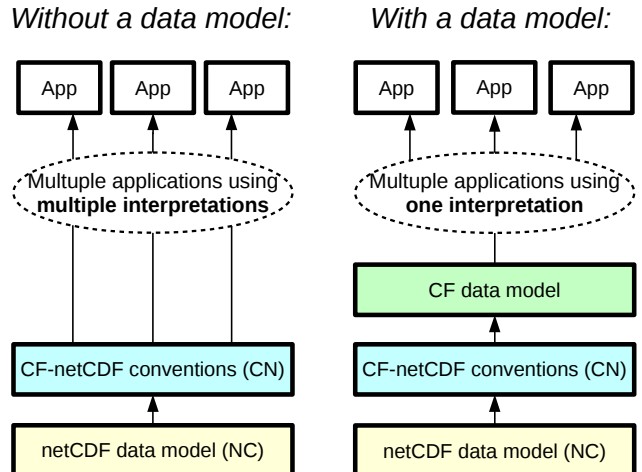

**Figure 1.** The benefits of having a CF data model. The CF-netCDF conventions (CN) rely on the netCDF data model (NC) and, at present, a software application is forced to make its own interpretation of the CF-netCDF conventions—an interpretation that is likely to be different from that of other applications. The aim of this paper is to propose a comprehensive CF data model that provides a single interpretation of the conventions, thereby facilitating compatibility across applications that adopt it.

variety of software implementations, as there is still considerable flexibility in mapping the data model elements onto the data objects needed for a particular application.

## 1.1  Design criteria for the CF data model

The primary requirement of the data model is that it should be able to describe all existing and conceivable CF-compliant datasets. If we have been successful, then software libraries that adopt our CF data model in constructing their internal data

structures will be able to represent and manipulate any CF-compliant dataset.

For our data model, we define a minimal set of elements that are sufficient for accommodating all aspects of the CF conventions. We restrict the elements of the data model to those that are explicitly mentioned in CF, but our data model elements do not have to be irreducible in that a data model element could describe more than one CF entity. For example, in CF, coordinates and coordinate bounds are distinct entities, but coordinate bounds cannot exist without coordinates. Therefore it makes sense

in our data model to group them into a single element.

Similarly, while it is possible to introduce additional elements not presently needed or used by CF, we believe this would not be desirable because it would increase the likelihood of the data model becoming outdated or inconsistent with future versions of CF.

The CF data model should also be independent of the encoding, meaning that it should not be constrained by the parts of the

CF conventions which describe explicitly how to store (i.e. encode) metadata in a netCDF file. the virtue of this is that should netCDF ever fail to meet the community needs, we shall already have set the groundwork for applying CF to other file formats.



**Table 1.** The UML class associations used in this paper. See figure A1 for a worked example.

| UML association | Description |
| --- | --- |
| class-B → class-A | Class-B is a special kind of class-A |
| class-E / class-D | Class-D is a special kind of class-E (but class-E is not shown on the diagram) |
| class-B ◆→ class-C | Instances of class-C can be included in an instance of class-B but cannot exist independently |
| class-B ◇→ class-F | Instances of class-F can be included in an instance of class-B or can exist independently |
| class-B → class-D | An instance of class-B is associated with an instance of class-D but class-D is independent of class-B |

## 1.2 Layout of the paper

In section 3 we introduce the key elements of the CF conventions and describe how they are encoded in netCDF files. The relationships between the elements of the CF conventions and our proposed CF data model are described in section 4. This data model is compared with other data models in section 5 and a software implementation is presented in section 6. A summary and conclusions are given in section 7.

## 2 The netCDF data model

The CF conventions have been developed for use with netCDF files following the netCDF "classic" data model (the yellow layer in figure 1). A brief summary of this explicit data model is useful since the CF conventions can not be described without reference to elements of netCDF.

The netCDF classic data model is described using Unified Modelling Language (UML) in figure 2. UML provides a standard way to visualise the components of a system and how they relate to each other. In UML, different styles of arrow denote different types of relationship (table 1). Appendix A provides a primer on the subset of UML used in this paper, and is recommended for readers new to this style of diagram.

NetCDF classic files contain data in named variables, which can be single numbers (with no dimensions), one-dimensional arrays (vectors), or multidimensional arrays, and the dimensions are declared by name in the file. Variables can be of integer, floating point or character data types. Variables may have attributes, of any data type, attached. Attributes can have a single value or consist of a one-dimensional array. NetCDF files also have "global" file attributes which provide information about the dataset as a whole. NetCDF library software has functions to define dimensions, variables and attributes, and write and read data.

It is important to appreciate that netCDF itself has no other semantics; for example, while coordinates can be stored in variables and described by attributes, the meanings of these variables and attributes and relationships between them and the variables containing data are not defined by netCDF. NetCDF makes no prescriptions or restrictions regarding the type of



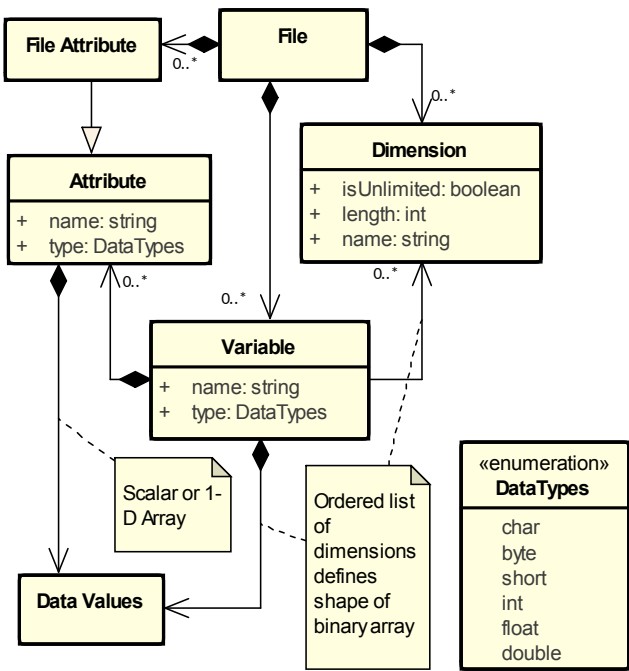

**Figure 2.** Key components of the netCDF classic data model (corresponding to the yellow "NC" layer in figure 1) described using UML (appendix A). Files consist of global attributes, dimensions and variables. Variables contain attributes and data, and attributes also contain data. Variables, attributes and dimensions all contain properties, such as a "name" which identifies them in the file. A data array has a data type for all of its elements (e.g. "double" for 64-bit floating point numbers).

metadata which may be stored in the simple data structures that it offers. This flexibility is intended to provide scope for users
and scientific disciplines to develop their own conventions for encoding semantics so that datasets are sufficiently described by those who create them and that they remain valid for those who store and use them. CF is an example of this.

The original classic netCDF data model has been "enhanced" with the addition of several new features, including the ability to organise variables in hierarchical groups. Here we adopt only one of the new features: we regard the character string as a data type, whereas the classic model treats strings as arrays of individual characters. Logically these treatments are equivalent,
but because strings are easier to manipulate in software codes, it is very likely that they will become a part of CF in the future.

## 3  The CF conventions

In this section we briefly describe how the most important of the CF conventions are encoded in netCDF files. We do not consider any conventions accepted after version 1.6. The comprehensive definition of CF, which includes many extra details, can be found on the CF website (http://cfconventions.org). A netCDF file that encodes an example of each aspect of the
110 conventions that we describe is shown in figure 3. This example file is presented in Common Data Language (CDL, Rew et al. (1997))—a human-readable notation for netCDF data that is easily produced by netCDF library software.



In order to reduce storage occupied by netCDF files, the CF conventions provide for lossy packing of data values and non-lossy compression eliminating missing data values. Although practically valuable, these mechanisms do not affect our conceptual data model and so we have chosen to describe them in appendix B rather than in this section.

## 3.1 Conventions from the netCDF user guide and COARDS

Unidata provides a netCDF user guide (NUG) (Rew et al., 1997), in which they propose some netCDF conventions. CF makes use of these conventions, so we regard them as part of the CF conventions as well (the blue layer in figure 1), and they are illustrated in figure 3. A one-dimensional variable that has the same name as its dimension ("z", "x" and "y" in figure 3) is regarded as a coordinate variable. Since CF introduces other types of variable for coordinate data, the kind defined in the netCDF user guide is sometimes referred to as a coordinate variable "in the NUG sense". We describe the various kinds in more detail later (section 3.3). The netCDF user guide proposes a number of conventional attributes; some of these are explicitly included in CF, which also contains a statement that it is always allowable to make use of attributes that are not standardised by CF. Some of the Unidata attributes recognised by CF contain scientific metadata e.g. "source", for the provenance of the data, and the "units" of data values (further discussed in section 3.8). Others of them concern the encoding in netCDF files, e.g. "Conventions", stating the netCDF conventions to which the file adheres (line 73 of figure 3) and the specification of missing data values with "_FillValue" (line 53 of figure 3, indicating that values of $-10^{30}$ correspond to missing data values).

When originally conceived, CF was an extension of the pre-existing COARDS (Cooperative Ocean/Atmosphere Research Data Service) netCDF conventions (http://www.ferret.noaa.gov/noaa_coop/coop_cdf_profile.html). For the sake of backward compatibility of datasets, although CF is now much more comprehensive and flexible than COARDS, CF explicitly upholds some COARDS conventions, which we therefore regard also as part of CF.

## 3.2 The data and the domain

The overarching purpose of the conventions is to provide conforming datasets with sufficient metadata that they are self-describing, in the sense that each variable in the file has an associated description of what it represents, and that each value can be located (usually in space and time). To meet this objective, we define a data variable $V$ (which might, for example, represent air temperature), over a domain $d$,

$$V = V(d) \tag{1}$$

where $d$ represents a set of discrete "locations" in what generally would be a multidimensional space, either in the real world or in a model's simulated world. Thus $V$ is a function of all its independent dimensions. For example, a variable that is a function of physical location alone would have a three-dimensional discretised domain,

$$d = d(z, y, x) \tag{2}$$

comprising discretised axes of height ($z$), latitude ($y$) and longitude ($x$) (figure 4). In CF, the domain may have fewer than three spatial axes, and it may also have any number of non-spatial axes, as in the common case of a variable that is a function





```
netcdf example_file {                                                                    // 1
dimensions:                                                                              // 2
    z = 20 ;                                                        // Domain axis construct  3
    y = 110 ;                                                       // Domain axis construct  4
    x = 106 ;                                                       // Domain axis construct  5
    bounds = 2 ;                                                                         // 6
variables:                                                                               // 7
    double t ;                                      // Domain axis and dimension coordinate construct  8
        t:standard_name = "time" ;                                                       // 9
        t:units = "days since 2016-12-01" ;                                              // 10
        t:calendar = "gregorian" ;                                                       // 11
        t:bounds = "t_bounds" ;                                                          // 12
    double z(z) ;                                   // Dimension coordinate and domain ancillary construct 13
        z:standard_name = "atmosphere_sigma_coordinate" ;                                // 14
        z:positive = "down" ;                                                            // 15
        z:units = "1" ;                                                                  // 16
        z:formula_terms = "sigma: z ps: PS ptop: PTOP" ;        // Coordinate reference construct 17
        z:bounds = "z_bounds" ;                                                          // 18
    double y(y) ;                                               // Dimension coordinate construct 19
        y:standard_name = "projection_y_coordinate" ;                                    // 20
        y:units = "km" ;                                                                 // 21
        y:bounds = "y_bounds" ;                                                          // 22
    double x(x) ;                                               // Dimension coordinate construct 23
        x:standard_name = "projection_x_coordinate" ;                                    // 24
        x:units = "km" ;                                                                 // 25
        x:bounds = "x_bounds" ;                                                          // 26
    double lon(y, x) ;                                         // Auxiliary coordinate construct 27
        lon:standard_name = "longitude" ;                                                // 28
        lon:units = "degrees_east" ;                                                     // 29
    double lat(y, x) ;                                         // Auxiliary coordinate construct 30
        lat:standard_name = "latitude" ;                                                 // 31
        lat:units = "degrees_north" ;                                                    // 32
    double t_bounds(bounds) ;                           // Part of a dimension coordinate construct 33
    double z_bounds(z, bounds) ;        // Part of a dimension coordinate and domain ancillary construct 34
        z_bounds:formula_terms = "sigma: z_bounds ps: PS ptop: PTOP" ;                   // 35
    double y_bounds(y, bounds) ;                        // Part of a dimension coordinate construct 36
    double x_bounds(x, bounds) ;                        // Part of a dimension coordinate construct 37
    double cell_area(y, x) ;                                        // Cell measures construct 38
        cell_area:standard_name = "area" ;                                               // 39
        cell_area:units = "m2" ;                                                         // 40
    char lambert_conformal ;                                   // Coordinate reference construct 41
        lambert_conformal:grid_mapping_name = "lambert_conformal_conic" ;                // 42
        lambert_conformal:standard_parallel = 25. ;                                      // 43
        lambert_conformal:longitude_of_central_meridian = 265. ;                         // 44
        lambert_conformal:latitude_of_projection_origin = 25. ;                          // 45
    double PS(y, x) ;                                             // Domain ancillary construct 46
        PS:standard_name = "surface_air_pressure" ;                                      // 47
        PS:units = "Pa" ;                                                                // 48
    double PTOP(y, x) ;                                           // Domain ancillary construct 49
        PTOP:standard_name = "air_pressure" ;                                            // 50
        PTOP:units = "Pa" ;                                                              // 51
    double temp(z, y, x) ;                                                // Field construct 52
        temp:_FillValue = -1.0e30 ;                                                      // 53
        temp:standard_name = "air_temperature" ;                                         // 54
        temp:units = "K" ;                                                               // 55
        temp:cell_methods = "t: mean (interval: 1 day)" ;         // Cell method construct 56
        temp:coordinates = "t lat lon" ;                                                 // 57
        temp:cell_measures = "area: cell_area" ;                                         // 58
        temp:grid_mapping = "lambert_conformal" ;                                        // 59
        temp:ancillary_variables = "temp_error_limit" ;                                  // 60
    double total_wv(y, x) ;                                               // Field construct 61
        total_wv:standard_name = "atmosphere_mass_content_of_water_vapor" ;              // 62
        total_wv:units = "kg m-2" ;                                                      // 63
        total_wv:cell_methods = "t: maximum" ;                    // Cell method construct 64
        total_wv:coordinates = "t lat lon" ;                                             // 65
        total_wv:cell_measures = "area: cell_area" ;                                     // 66
        total_wv:grid_mapping = "lambert_conformal" ;                                    // 67
    double temp_error_limit(z, y, x) ;                            // Field ancillary construct 68
        temp_error_limit:standard_name = "air_temperature standard_error" ;              // 69
        temp_error_limit:units = "K" ;                                                   // 70
                                                                                         // 71
// global attributes:                                                                    // 72
        :Conventions = "CF-1.6" ;                                                        // 73
        :source = "climate model" ;                                                      // 74
}                                                                                        // 75
```

**Figure 3.** A CDL representation of the CF-netCDF file used for examples in section 3 and to demonstrate the software implementation in section 6. Data values have been omitted for brevity. Each line has a comment on the right hand side (beginning with //) that gives the line number and notes the CF data model constructs (section 4) which correspond to netCDF dimensions, variables and attributes.





**Figure 4.** An example domain defined by three dimensions, one of which is single-valued (height).

time ($t$). A CF-netCDF file may contain $N$ data variables and $M$ domains, where $M \leq N$. Conversely, this means that a given domain may have one or more data variables defined at each of its locations. For instance, there could be values for both air temperature and relative humidity at each location in the domain $d(z, y, x)$.

In CF-netCDF, the values and the description of $V$ are stored in a netCDF variable, called a "data variable". The concept of a domain is not mentioned in the CF conventions, because it does not correspond to any single entity in the netCDF file. The domain is stored in a number of other variables and attributes that are linked to the data variable in various ways defined by the conventions. For instance, "temp" is a data variable (line 52 in figure 3) with a four-dimensional domain. Its $t$, $z$, $y$ and $x$ dimensions each have an associated coordinate variable specifying the location at each point along the dimension (see section 3.3 for details).

Within a CF-netCDF file, dimensions and coordinate variables may be used in the definition of multiple domains, thus reducing redundancy. In our example file, "total_wv" is a data variable containing the vertical integral of atmospheric water vapour (line 61 in figure 3) that has a different domain than the "temp" data variable. Its $t$, $y$ and $x$ dimensions, and their coordinates (see section 3.3), are identical with those of "temp" so they need not be replicated, but it does not require the $z$ dimension.

### 3.3 Dimensions and Coordinates

NetCDF dimensions establish the size of the index space of data variables e.g. lines 3–5 in figure 3, which specify sizes of 106, 110 and 20 for the $x$, $y$ and $z$ dimensions respectively. Each point of the domain of "temp" is thus defined by a unique set of three indices $i = 0, \ldots, 105$, $j = 0, \ldots, 109$ and $k = 0, \ldots, 19$. NetCDF coordinate variables (in the NUG sense) supply the independent variables on which the data depend. Coordinate variables must be numeric and strictly monotonic, so that each element has a unique value. In our example we have three coordinate variables, with values $z(k)$, $y(j)$, $x(i)$. Each dimension, with its coordinate variable if it has one, constitutes an axis of the multidimensional space of the domain. The CF conventions quite often uses the word "axis" to refer to the physical interpretation of the dimensions of the data.

In many cases, each dimension of a domain can be fully described by a single strictly monotonic coordinate variable (e.g. time, height, latitude, longitude). However, for more complicated cases, such as parametric vertical coordinates (e.g. dimension-



less atmosphere sigma coordinates), CF provides a way to record how to compute from the original coordinates dimensional coordinates identifying the location of the data in physical space (in the case of sigma, the air pressure). This information is encoded with the "standard_name" and "formula_terms" attributes of a parametric coordinate variable e.g. lines 14 and 17

in figure 3. The standard_name attribute defines the formula for calculating the dimensional coordinates, which needs to be looked up in the CF conventions document, and the formula_terms attribute specifies the values of the formula's terms. The "atmosphere_sigma_coordinate" formula specified in figure 3 calculates air pressure from $\mathrm{ptop} + \mathrm{sigma} * (\mathrm{ps} - \mathrm{ptop})$ where the values of "ptop" (pressure at the top of the model), "sigma" (the dimensionless coordinates) and "ps" (surface pressure) are taken from the netCDF variables referenced by the formula_terms attribute.

CF also defines "auxiliary coordinate variables" to provide additional or alternative coordinate information. They can be string-valued, may contain missing values and are not necessarily monotonic. For example, we might like to associate the coordinates of a vertical axis with model level number as well as sigma coordinate, or to provide location information and station names for the points in a timeseries (as in figure 5). An auxiliary coordinate variable is encoded as a netCDF variable that is referenced by the "coordinates" attribute of a data variable and spans at least one of that data variable's dimensions, e.g.

lines 27, 30 and 57 in figure 3. Coordinate variables (in the NUG sense) and auxiliary coordinate variables (defined by CF) rely on different semantics, and the latter is not a special type of the former, even though they share many characteristics.

An important use of auxiliary coordinates is to supply latitude and longitude locations of each point when the horizontal axes of a grid are themselves not latitude and longitude (e.g. if they refer to a rotated north pole, or are based on a map projection, as is the case for $x$ and $y$ in the example of figure 3, and sketched in figure 6). For this case, the latitude and longitude auxiliary

coordinate variables are two-dimensional and can be used to indirectly locate a point horizontally, so that $V = V(y_j, x_i)$ with longitude $= \mathrm{longitude}(y_j, x_i)$, and latitude $= \mathrm{latitude}(y_j, x_i)$, as in lines 27 and 30 of figure 3. Given this information, the data can be located in space by generic applications even if they are ignorant of the rules used to construct the projection. However, CF also provides for information about the grid construction to be included in a netCDF file by defining a "grid mapping" variable, which is referenced by the "grid_mapping" attribute of a data variable e.g. lines 41 and 59 in figure 3.

Some axes have only a single coordinate value. Regrettably, single-valued coordinates are often omitted from metadata, although they are very useful—for example, the time information for a field sampled at a single time, say 12:15 on 14th July 2015, or the level of a single-level field e.g. air temperature at a height of 1.5 m (figure 4). For convenience in storing single-valued coordinates, CF defines a third type of variable containing coordinate data, namely a "scalar coordinate variable", which requires less netCDF machinery than a dimension of size unity. It is a zero-dimensional netCDF variable that is referenced by

the "coordinates" attribute of a data variable e.g. lines 8 and 57 in figure 3.

Calendar time in CF (year, month, day, hour, minute, second) is encoded with units "time unit since reference date-time" (e.g. line 9 in figure 3). The encoded coordinates are the elapsed times since the reference, and as such are useful for computing time differences. CF does not use strings for time because they cannot be used for such computations, are inconvenient to standardise, take more storage space and cannot always be ordered monotonically. The encoding depends on the calendar (e.g.

line 10 in figure 3), which defines the permitted values of the reference date (year, month and day). The format of the units string conforms to udunits syntax (section 3.8), but the udunits software supports only the real-world Julian/Gregorian calendar,





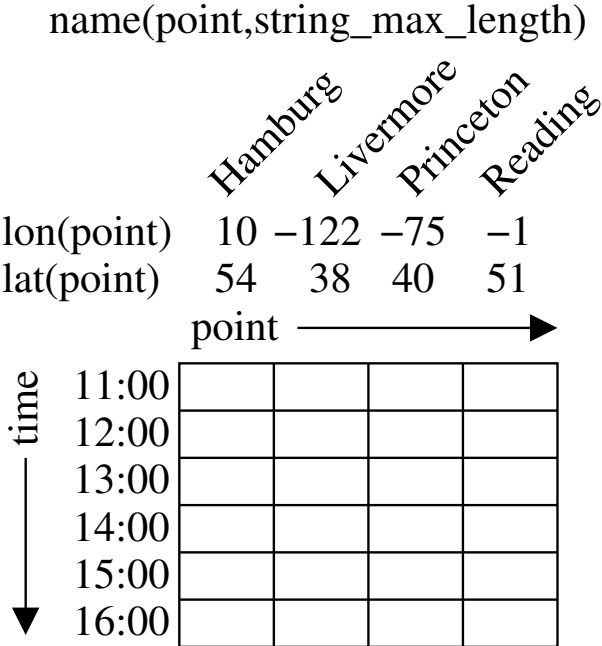

**Figure 5.** Auxiliary coordinate variables store "alternative" coordinates for dimensions.

and hence is not sufficient for use with CF, which recognises a wide selection of calendars, including those for climate models and palaeoclimate. For instance, 31st August 2003 is a valid date in the real-world Gregorian calendar, but not in the "360-day" calendar, which has twelve 30-day months. In the Gregorian calendar, 12:00 on 29th February 2000 is 36583.5 days since 0:00 on 1st January 1900, but it is 36058.5 days in the 360-day calendar.

### 3.4 Discrete axes and sampling geometries

A "discrete axis" is one which is not associated with any "continuous" coordinate or auxiliary coordinate variables. A variable is continuous along an axis if it makes physical sense to interpolate along that axis between its values. If that is not the case then either there are no coordinate values or the coordinate values are discrete indices, whose order may or may not be meaningful. Consider, for example, an ensemble of model experiments, each of which produces a data variable $V(t,z,y,x)$ containing air pressure as a function of time and spatial location. It may be convenient to combine the data variables into a single variable $V(e,t,z,y,x)$, where $e$ is the ensemble dimension, defining a discrete axis of ensemble members. The members could be identified by a numeric monotonic coordinate variable with the same name as the dimension containing a member number. Alternatively, it is common for them to be identified by one or more strings, which we could store in auxiliary coordinate variables with the ensemble dimension containing, for instance, model names or experiment names. It usually would not make sense to interpolate between ensemble members, and their order may be immaterial.



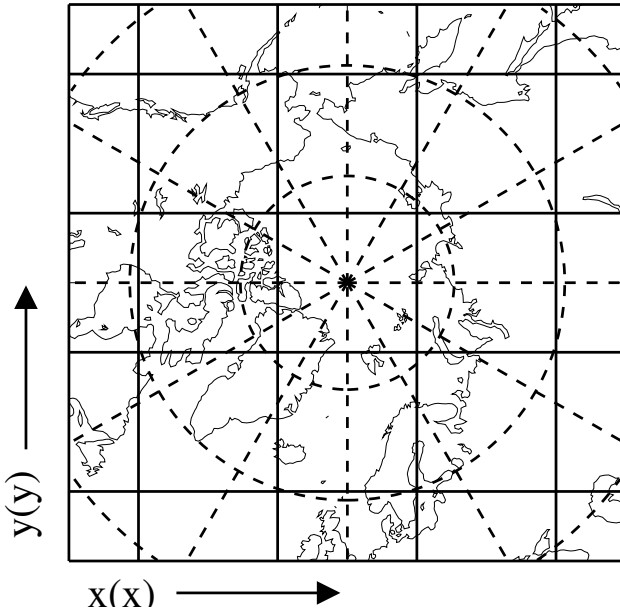

**Figure 6.** For grid axes based on a map projection, two-dimensional auxiliary coordinate variables must be used to store longitude and latitude values for each location (latitude–longitude lines dashed, grid-lines solid).

An important use of discrete axes in CF is to store data from a collection of "discrete sampling geometries" (DSGs) in a single data variable. In a DSG, the data has a lower dimensionality than the space–time domain, because it applies to a point or path within the domain. For example, a collection of timeseries of surface air temperature at meteorological stations can be stored in a two-dimensional data variable (figure 5) with a dimension that is the discrete axis for the stations (the "point" axis) and a dimension for the timeseries values (the "time" axis). Auxiliary coordinate variables for the discrete axis can be used to provide location information and station names (figure 5). Although the stations are physically located in two or three spatial dimensions, these have been combined into the one discrete axis. Other examples of DSGs are a vertical profile (variation along a vertical axis at a fixed time and spatial location) and a trajectory (variation along a path through space as a function of time). A DSG may also be called a "feature" and the type of DSG is called its "feature type". The feature type (point, timeseries, trajectory, profile, etc.) describes the DSG and specifies the dimensionality of the data within the space–time domain. Prior to the introduction of DSGs in version 1.6 of CF, the concept of DSG features was implicit in the sense that they could be inferred from the existence of a discrete axis and well-defined coordinate and auxiliary coordinate variables, but they were not formally described. In general the netCDF file attribute "featureType" specifies the DSG feature type for every data variable in the file; but in the cases where the featureType attribute is permitted to be missing, the feature type may be inferred from the dimensions and space–time coordinates alone.

Many DSGs many be stored in one file, but they might have different coordinates e.g. each timeseries might have its own set of sampling times (different days or hours of observation), or each profile might have its own set of vertical levels (e.g.





air pressure reported by radiosondes). If each feature in a large collection is stored as an individual data variable with its own

dimensions and coordinate, the file will be cumbersome. If they are combined into a single data variable, a one-dimensional

coordinate variable would need to contain the union of all the coordinates (times, levels, etc.) required, and the dimension of

the combined data variable might be much larger than needed for the available data, containing a lot of missing data elements.

As an alternative CF provides three other methods for storing collections of data on DSGs, all intended to allow data with

different dimensionality to be stored in a single data variable without wasting so much space. In the "incomplete multidi-

mensional array" representation, the dimension required for the longest feature is used for all features, so that the shorter

features must be padded with missing values; this sacrifices storage space to achieve simplicity for reading and writing. The

"contiguous ragged array" and "indexed ragged array" representations eliminate the need for padding and thus reduce further

the storage required, but are more complex to pack and unpack. In the former case, each feature in the collection occupies a

contiguous block, requiring the size of each feature to be known at the time that it is created. In the latter case, the values of

each feature in the collection are interleaved. This representation can therefore be used for real-time data streams that contain

reports from many sources, with the data being written as it arrives. The ragged array representations are described in more

detail in appendix B.

Because these storage methods were introduced (in CF version 1.6) at the same time as the recognition and definition of

feature types, the two are often thought of as belonging together, but this causes confusion. The featureType is metadata, and

it refers to the physical construction and interpretation of a DSG data variable. The three new storage mechanisms for DSGs

do not involve any new or distinct physical concepts.

### 3.5   Bounds and cells

It is often necessary to know the extent of a cell as well as the grid-point location, e.g. to calculate the area of a latitude-

longitude box or the thickness of a vertical layer. If cell bounds are not provided then there is no default assumption about

cell sizes (an application might reasonably assume that grid points are at the centres of non-overlapping cells, but that is not

required by CF).

CF provides a way to attach bounds variables to any variable containing coordinate data. A bounds variable has an extra

dimension to index the vertices of the cells. The simplest case is shown for a one-dimensional coordinate variable in figure 7.

In this case, the values $p(i)$ may be a series of successive time-instants, for example, midday on 6th and 7th of November,

bounded by $b(i, q)$ at midnight on 6th, 7th and 8th, with $q = 0, 1$. While the bounds for a one-dimensional coordinate variable

of dimension $(n)$ could often be stored in a vector of dimension $(n+1)$, CF uses $(n, 2)$ instead as shown because it is convenient

for use with the netCDF unlimited dimension, and because it allows cells to be non-contiguous or overlapping. The bounds can

be used to test contiguity; in the figure, cell $i$ and cell $i + 1$ are contiguous because $b(i + 1, 0) = b(i, 1)$. For multidimensional

auxiliary coordinate variables, such as the two-dimensional latitude and longitude variables illustrated above, we have to supply

the coordinates of each vertex of the polygon and contiguity can similarly be tested by coincidence of vertices. A bounds

variable is encoded as a netCDF variable that is referenced by the "bounds" attribute of a coordinate or auxiliary coordinate




variable and spans the same dimensions (as well as the extra dimension defining the number of vertices), e.g. lines 6, 22 and 36 in figure 3.

Some applications require information about the size, shape or location of the cells that cannot be deduced without specialist knowledge which is not guaranteed to be available. For example, in computing the mean of several cell values, it is often appropriate to "weight" the values by area, but for some grids (such as some types of spherical geodesic grid) the cell perimeter is not uniquely defined by its vertices and so the area can not be inferred from the available information. For this case, CF provides cell measures variables which contain such information and are encoded as netCDF variables which are referenced by the "cell_measures" attribute of a data variable and span a subset of the data variable's dimensions, e.g. lines 38 and 58 in figure 3.

### 3.6 Variation within cells

CF describes variation within cells by use of "cell methods". By default, it is assumed that intensive quantities apply at grid-points e.g. temperature values apply at the spatial points and instants of time specified by their coordinates, while extensive quantities apply to the entire grid-cell e.g. a precipitation amount (kg m$^{-2}$) is an accumulation in time. The method may be different for each axis e.g. precipitation amount is intensive in space even though it is extensive in time. Because the default is not always obvious, it is recommended that the method be stated explicitly for every axis. Non-default methods include operations such as mean, maximum, minimum and standard deviation. A zonal-mean variable, for instance, has a cell methods attribute that specifies it is a mean over longitude. A timeseries of daily maximum values has a cell methods indicating that the values are maxima within their cells in time. The operations recorded by cell methods might affect more than one axis at once, e.g. for the cell methods necessary to describe maximum of the ocean meridional overturning streamfunction within a depth–latitude cell.

By default, the method for horizontal cells is assumed to have been evaluated over the entire area of the cell. It is, however, possible to limit consideration to only a portion of a cell, e.g. to record that values apply only to the fractions of cells which are land (as opposed to sea).

A further use of cell methods is to characterise climatological statistics where a series of data points represent sets of subintervals which are not contiguous. There are three kinds to consider:

1. Corresponding portions of the annual cycle in a set of years, e.g. decadal averages for January.

2. Corresponding portions of a range of days, e.g. the average diurnal cycle in April 1997.

3. Both at once, e.g. the average winter daily minimum temperature from the years 1961 to 1990.

In the latter example the bounds are 0Z 1st December 1961 (beginning of the first day of the first winter) and 0Z 1st March 1991 (end of the last day of the last winter), and the cell methods indicates the values are a minimum within days, a mean over a season and a mean over years.

Cell methods are encoded in the "cell_methods" attribute of a data variable, e.g. line 56 in figure 3. In this example, the cell method "t: mean (interval: 1 day)" specifies that data values are means over the time dimension (i.e. temporal averages)



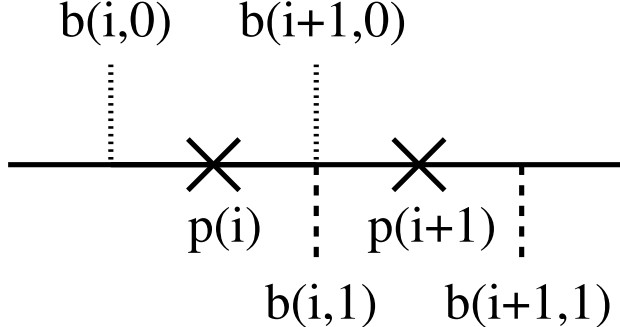

**Figure 7.** A one-dimensional coordinate variable with grid-points $p(i)$ and cell boundaries $b(i,q)$, $i = 1, \ldots, n$; $q = 0, 1$.

calculated from daily samples. Note that this netCDF file does not contain a dimension for time, but it is sufficient that one is implied by the time scalar coordinate variable on line 8.

### 3.7  Ancillary data

When metadata to describe the data depends on location within the domain, it is stored in independent variables called ancillary data variables. For example, each value of an array of instrument data may have associated measures of uncertainty, or of

the status of the recording instrument. An ancillary data variable is encoded as a netCDF variable that is referenced by the "ancillary_variables" attribute of a data variable and spans a subset of the data variable's dimensions, e.g. lines 60 and 68 in figure 3.

### 3.8  Units and standard name

A range of attributes is available, introduced by the netCDF user guide or CF, providing metadata for interpreting the values of

individual variables or about the dataset as a whole. In this section we discuss the two most important of these.

CF requires all variables with values (data variables, coordinate variables, etc.) to have units unless they contain dimension-less numbers or cell boundary values. The units are specified by a string attribute (e.g. lines 9 and 55 of figure 3), formatted according to the Unidata udunits conventions (Emmerson, 2007), which support many possible units and varieties of syntax e.g. metre, meter, meters, m, km, cm, second, s, kelvin, K, Pa, W m-2, W/m^2, kg/m2/s, 1 (or any number). Many non-SI

units are also supported by udunits e.g. degree, degree_north, degree_N, percent (equivalent to 0.01), ppm (equivalent to 1e-6), mbar, mile, degC, degF, hours, day.

For systematic identification of the physical quantity contained in variables, CF defines a "standard name" string attribute (e.g. lines 28, 54 and 62 of figure 3), with permissible values listed in the standard name table (http://cfconventions.org/ standard-names.html), which includes precise definitions. The standard name table is managed by a community process and is

continually expanding—version 44 of the table, released in May 2017, contains 2847 standard names.



CF also upholds the use of the "long name" defined by the netCDF user guide, but this is ad-hoc. In contrast, the CF standard names are consistently constructed and documented. As CF is applicable to many areas of geoscience, the standard names have to be more self-explanatory and informative than would suffice for any one area. For instance, there is no name for plain "potential temperature", since we have to distinguish air potential temperature and sea water potential temperature. Standard

names are often longer than the terms familiarly used by the experts in particular discipline, because they answer the question, "What does this mean?", rather than the question, "What do you call this?". For example, the quantity often called "precipitable water" by meteorologists has the standard name of atmosphere_mass_content_of_water_vapor. Standard names have a detailed description which further defines parts of the name; for example, the description of the standard name land_ice_calving_rate notes that "land ice" means glaciers, ice-caps and ice-sheets resting on bedrock and the land ice calving rate is the rate at which

ice is lost per unit area through calving into the ocean. Each standard name also implies particular physical dimensions (mass, length, time and other dimensions corresponding to SI base units, expressed as a "canonical unit"); for example, large-scale rainfall amount (canonical unit $kg\ m^{-2}$), large-scale rainfall flux ($kg\ m^{-2}\ s^{-1}$) and large-scale rainfall rate ($m\ s^{-1}$) are all different in CF, although they might all be vaguely referred to as "large-scale rain".

Standard names have been defined for both more general and more specific quantities, for different applications e.g. ocean_-

mixed_layer_thickness and ocean_mixed_layer_thickness_defined_by_temperature. Some standard names require the existence of additional metadata and/or constraints on the values of the variables with which they are associated. For example, the standard name of downwelling_radiance_per_unit_wavelength_in_air requires there to be a coordinate variable storing the radiation wavelength.

The CF conventions use size one or scalar coordinate variables (section 3.3) and the cell_methods attribute (section 3.6) to

describe some aspects of a variable, and this means standard names do not always correspond to identities of variables in other file formats. For instance, to describe the time-mean air temperature at 1.5 m above the ground, air_temperature alone is the standard name; "time-mean" is described by cell_methods and the height as a coordinate.

## 4   A CF Data Model

The aspects of the CF conventions discussed in section 3 (i.e. CF-netCDF elements) are listed in table 2 and shown (in blue)

with their interrelationships in UML (appendix A) in figure 8. The CF-netCDF elements and the relationships among them could be regarded as defining a data model for CF (rather like that described in section 5.2), but that would not be the one we propose. Our CF data model (the green layer in figure 1) has been derived from these CF-netCDF elements and relationships with the aims of removing aspects specific to the netCDF encoding, and reducing the number of elements, whilst retaining the ability to describe the CF conventions fully. The elements of our CF data model are called "constructs", a term chosen

to differentiate from the CF-netCDF elements previously defined and to be programming language-neutral (i.e. as opposed to "object" or "structure"). In this section, we relate the constructs to the CF-netCDF elements of section 3, which in turn relates the CF-netCDF elements to the components of netCDF files. To clarify these connections, the example netCDF file shown in figure 3 indicates how its elements relate to the constructs of the CF data model.



**Table 2.** The elements of the CF-netCDF conventions, a brief description of each and the section in which it is described in more detail. The relationships to netCDF entities are shown in figure 8.

| CF-netCDF element | Description | Section |
|---|---|---|
| Data variable | Scientific data discretised within a domain | 3.2 |
| Dimension | Independent axis of the domain | 3.3 |
| Coordinate variable | Unique coordinates for a single axis | 3.3 |
| Auxiliary coordinate variable | Additional or alternative coordinates for any axes | 3.3 |
| Scalar coordinate variable | Coordinate for an implied size one axis | 3.3 |
| Grid mapping variable | Horizontal coordinate system | 3.3 |
| Boundary variable | Cell vertices | 3.5 |
| Cell measure variable | Cell areas or volumes | 3.5 |
| Ancillary data variable | Metadata that depends on the domain | 3.7 |
| Formula terms attribute | Vertical coordinate system | 3.3 |
| Feature type attribute | Characteristics of discrete sampling geometry | 3.4 |
| Cell methods attribute | Description of variation within cells | 3.6 |

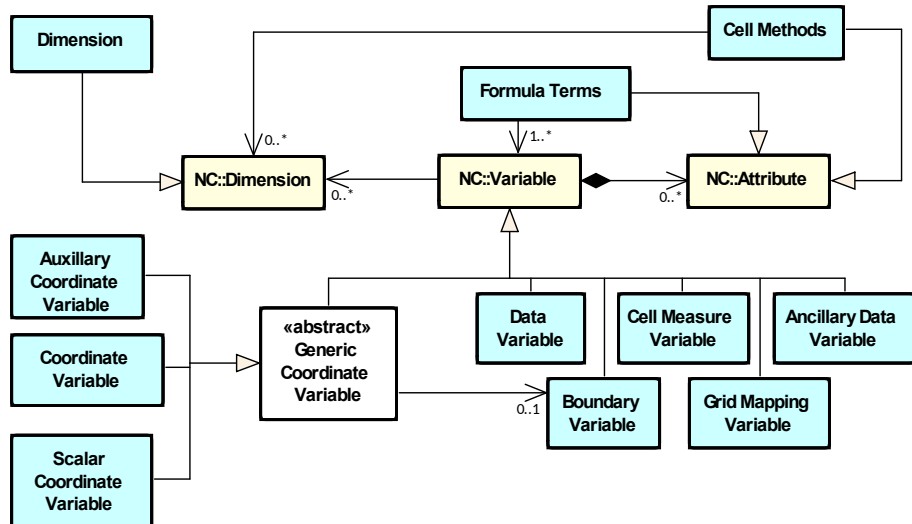

**Figure 8.** The relationships between CF-netCDF elements (corresponding to the blue "CN" layer in figure 1) and their corresponding netCDF variables, dimensions and attributes (the yellow "NC" layer in figure 1) described using UML (appendix A). It is useful to define an abstract generic coordinate variable that can be used to refer to coordinates when the their type (coordinate, auxiliary or scalar coordinate variable) is not an issue.



**Table 3.** The constructs of the CF data model, a brief description of each and the section in which it is described in more detail. The relationships between the constructs and CF-netCDF elements are shown in in figures 9–11.

| CF construct | Description | Section |
|---|---|---|
| Field | Scientific data discretised within a domain | 4.1 |
| Domain axis | Independent axes of the domain | 4.2 |
| Dimension coordinate | Domain cell locations | 4.3 |
| Auxiliary coordinate | Domain cell locations | 4.3 |
| Coordinate reference | Domain coordinate systems | 4.4 |
| Domain ancillary | Cell locations in alternative coordinate systems | 4.5 |
| Cell measure | Domain cell size or shape | 4.6 |
| Field ancillary | Ancillary metadata which varies within the domain | 4.7 |
| Cell method | Describes how data represents variation within cells | 4.8 |

### 4.1 The field construct

The field construct is central to the CF data model, and includes all the other constructs (figure 9). A field corresponds to a CF-netCDF data variable with all of its metadata. All CF-netCDF elements are mapped to some element of the CF field construct, as we describe in following subsections, and the field constructs completely contain all the data and metadata which can be extracted from the file using the CF conventions.

The field construct consists of a data array and the definition of its domain (i.e. $V(d)$ in equation 1), ancillary metadata
fields defined over the same domain, and cell methods constructs to describe how the cell values represent the variation of the physical quantity within the cells of the domain (figure 9). Because the CF conventions do not mention the concept of the domain, we do not regard it as a construct of the data model. Instead, the domain is defined collectively by various other constructs included in the field. All of the data model constructs are listed in table 3, which refers to the sections and figures in which they are fully described. All of the constructs contained by the field construct are optional (as indicated by "0..*" in
figure 9). The only component of the field which is mandatory is the data array.

The field construct also has optional properties to describe aspects of the data that are independent of the domain. These correspond to some netCDF attributes of variables (e.g. the units, long_name and standard_name, section 3.8), and some netCDF global file attributes (e.g. "history" and "institution"). We use the term "property", rather than "attribute", because not all CF-netCDF attributes are properties in this sense—some CF-netCDF attributes are used to point to (i.e. reference)
other netCDF variables and so only describe the data indirectly (e.g. the "coordinates" attribute, section 3.3), and others have structural functions in the CF-netCDF file (e.g. the "Conventions" attribute). In the CF data model, we consider that netCDF global file attributes apply to every data variable in the file, except where they are superseded by netCDF data variable attributes with the same name. This interpretation of global file attributes is not stated in the CF conventions but for our data model it is necessary because there is no notion of a file. Hence metadata stored in attributes of the file as a whole have to be transferred



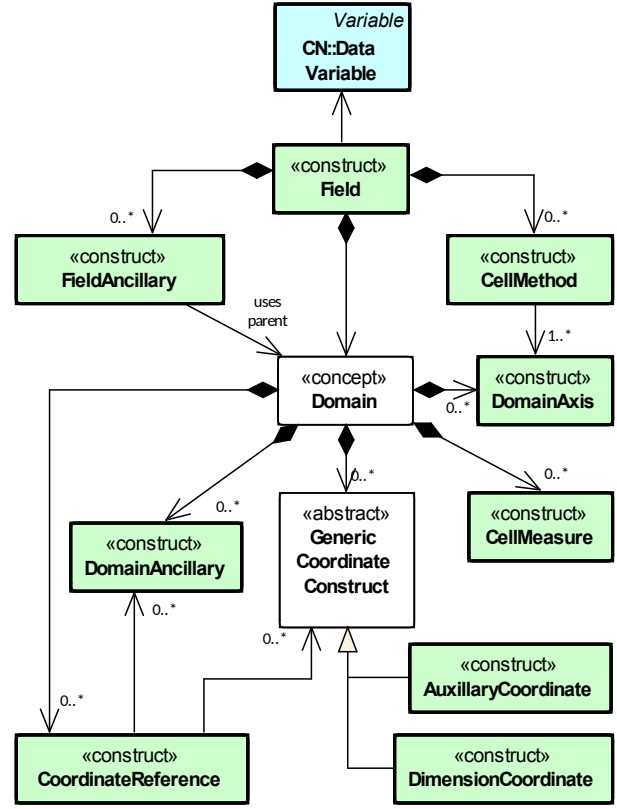

**Figure 9.** The nine constructs of the CF data model (corresponding to the green layer in figure 1) described using UML (appendix A). In this and all other CF data model diagrams, the CF data model constructs are labelled with the "construct" stereotype to distinguish them from other data model elements which appear in other diagrams as green boxes with no label. The field construct corresponds to a CF-netCDF data variable. Relationships between other constructs and CF-netCDF are given in figures 10 and 11. The domain provides the linkage between the field construct and the constructs which describe measurement locations and cell properties. It is not a construct of the data model (see section 4.1), but is an abstract concept that is useful for understanding the data model. Similarly, it is useful to define an abstract generic coordinate construct that can be used to refer to coordinates when the their type (dimension or auxiliary coordinate construct) is not an issue.

to the field construct. If present, the global file attribute "featureType" applies to every data variable in the file with a discrete sampling geometry (section 3.4). Hence we regard "feature type" as a property of the field construct.

The standard_name property (section 3.8) constrains the units property (i.e. only certain units are consistent with each standard name) and in some cases also the dimensions that a data variable must have. These constraints, however, do not supply any further information—they are just for self-consistency. This is also the case for the feature type property, which

imposes some requirements on the axes the domain must have. Following our aim of constructing a minimal data model, we do not regard the standard name nor feature type as separate constructs within the field, because they do not depend on any other





construct for their interpretation. This is unlike a cell method, for instance, which depends on the data variable's dimensions for its interpretation.

## 4.2 Domain axis construct and the data array

A domain axis construct (figure 10) specifies the number of points along an independent axis of the domain. It comprises a positive integer representing the size of the axis. In CF-netCDF it is usually defined either by a netCDF dimension or by a scalar coordinate variable, which implies a domain axis of size one (section 3.3). The field construct's data array spans the domain axis constructs of the domain, with the optional exception of size one axes, because their presence makes no difference to the order of the elements. Hence the data array may be zero-dimensional (i.e. scalar) if there are no domain axis constructs

of size greater than one.

     When a collection of DSG features has been combined in a data variable using the incomplete orthogonal or ragged representations to save space, the axis size has to be inferred, but we regard this as an aspect of unpacking the data, rather than its conceptual description. In practice, the unpacked data array may be dominated by missing values (as could occur, for example, if all features in a collection of timeseries had no common time coordinates), in which case it may be preferable to view the

collection as if each DSG feature were a separate variable (section 3.4), each one corresponding to a different field construct.

## 4.3 Coordinates: dimension coordinate and auxiliary constructs

Coordinate constructs (figure 10) provide information which locate the cells of the domain and which depend on a subset of the domain axis constructs. As previously discussed, there are two distinct types of coordinate construct: a dimension coordinate construct provides monotonic numeric coordinates for a single domain axis, and an auxiliary coordinate construct provides any

type of coordinate information for one or more of the domain axes.

     In both cases, the coordinate construct consists of a data array of the coordinate values which spans a subset of the domain axis constructs, an optional array of cell bounds recording the extents of each cell, and properties to describe the coordinates (in the same sense as for the field construct). An array of cell bounds spans the same domain axes as its coordinate array, with the addition of an extra dimension whose size is that of the number of vertices of each cell. This extra dimension does not

correspond to a domain axis construct since it does not relate to an independent axis of the domain (for example, the "bounds" dimension defined at line 6 in figure 3 does not correspond to a domain axis construct). Note that, for climatological time axes, the bounds are interpreted in a special way indicated by the cell method constructs.

     The dimension coordinate construct is able to unambiguously describe cell locations because a domain axis can be associated with at most one dimension coordinate construct, whose data array values must all be non-missing and strictly monotonically

increasing or decreasing. They must also all be of the same numeric data type. If cell bounds are provided then each cell must have exactly two vertices. CF-netCDF coordinate variables and numeric scalar coordinate variables correspond to dimension coordinate constructs.

     Auxiliary coordinate constructs have to be used, instead of dimension coordinate constructs, when a single domain axis requires more then one set of coordinate values, when coordinate values are not numeric, strictly monotonic or contain missing





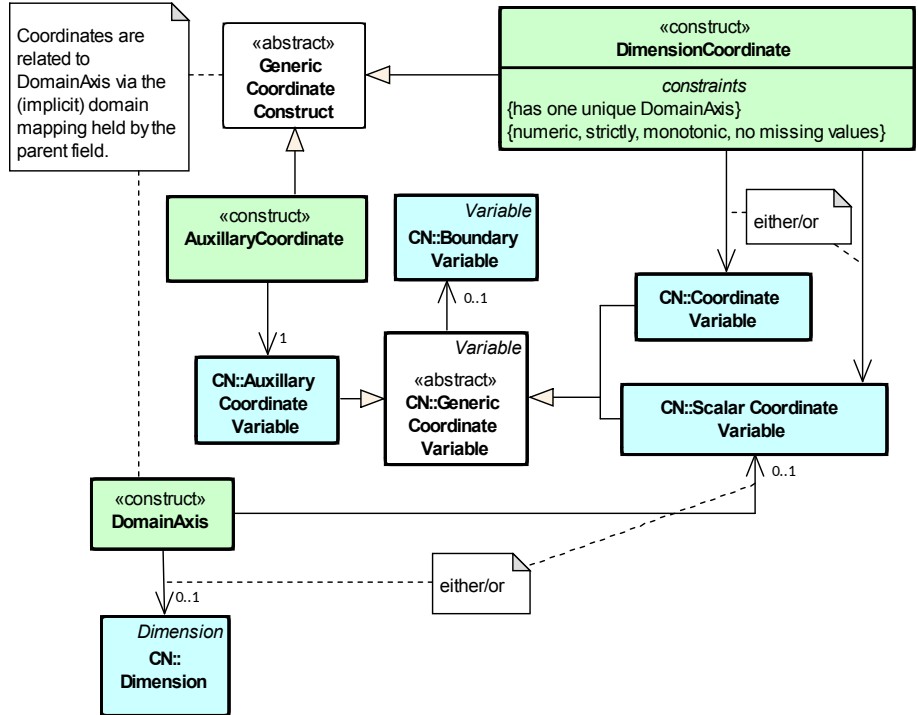

**Figure 10.** The relationship between domain axis, dimension coordinate and auxiliary coordinate constructs (sections 4.2 and 4.3) and CF-netCDF described using UML (appendix A). A dimension or auxiliary coordinate construct is defined by a CF-netCDF coordinate, scalar coordinate or auxiliary coordinate variable, and the associated CF-netCDF boundary variable if it exists. A generic coordinate construct spans one or more domain axis constructs, but the mapping of which ones is only held by the parent field construct.

values, or when they vary along more than one domain axis construct simultaneously. CF-netCDF auxiliary coordinate variables and non-numeric scalar coordinate variables correspond to auxiliary coordinate constructs.

    If a domain axis construct does not correspond to a continuous physical quantity then it is not necessary for it to be associated with a dimension coordinate construct. For example, this is the case for an axis that runs over ocean basins or area types, or for a domain axis that indexes a timeseries at scattered points. In such cases one-dimensional auxiliary coordinate constructs could be used to store coordinate values. These axes are discrete axes in CF-netCDF.

### 4.4 Coordinate reference construct

The domain may contain various coordinate systems, each of which is constructed from a subset of the dimension and auxiliary coordinate constructs. For example, the domain of a four-dimensional field construct may contain horizontal ($y$–$x$), vertical ($z$) and temporal coordinate ($t$) systems. There may be more than one of each of these, if there is more than one coordinate

construct applying to a particular spatiotemporal dimension (for example, there could be both latitude–longitude and $y$–$x$





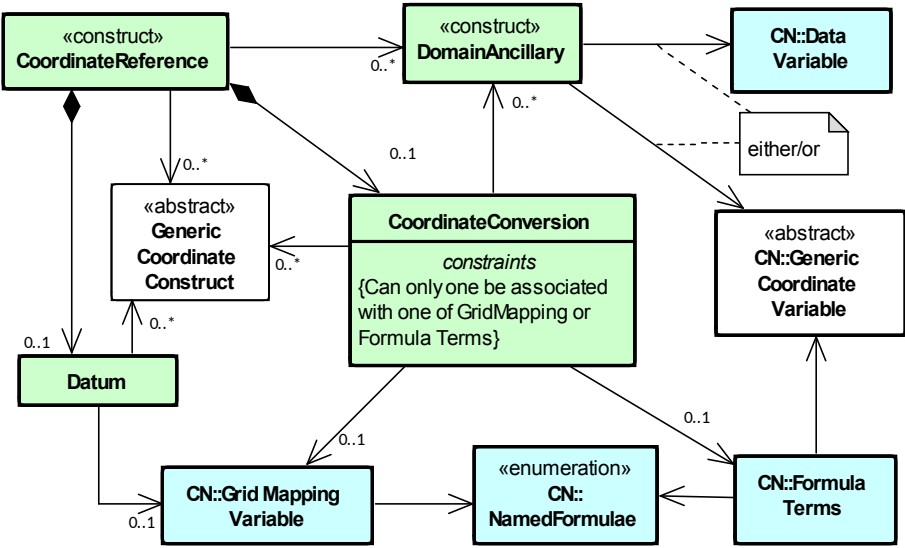

**Figure 11.** The relationship between coordinate reference and domain ancillary constructs (sections 4.4 and 4.5) and CF-netCDF described using UML (appendix A). A coordinate reference construct is defined either by a grid mapping variable, or a formula_terms attribute of a CF-netCDF coordinate variable. The coordinate reference construct is composed of generic coordinate constructs, a datum, and a coordinate conversion formula. The coordinate conversion formula is usually defined by a named formula in the CF conventions. A domain ancillary construct term of a coordinate conversion formula is defined by a CF-netCDF data variable or a CF-netCDF generic coordinate variable.

projection coordinate systems). In general, a coordinate system may be constructed implicitly from any subset of the coordinate constructs, yet a coordinate construct does not need to be explicitly or exclusively associated with any coordinate system.

A coordinate system of the field construct can be explicitly defined by a coordinate reference construct (figure 11) which relates the coordinate values of the coordinate system to locations in a planetary reference frame, and consists of

– the dimension coordinate and auxiliary coordinate constructs that define the coordinate system to which the coordinate reference construct applies. Note that the coordinate values are not relevant to the coordinate reference construct, only their properties.

     – A definition of a datum specifying the zeroes of the dimension and auxiliary coordinate constructs which define the coordinate system. The datum may be explicitly indicated via properties, or it may be implied by the metadata of the

contained dimension and auxiliary coordinate constructs. Note that the datum may contain the definition of a geophysical surface which corresponds to the zero of a vertical coordinate construct, and this may be required for both horizontal and vertical coordinate systems.

     – A coordinate conversion, which defines a formula for converting coordinate values taken from the dimension or auxiliary coordinate constructs to a different coordinate system. A term of the conversion formula can be a scalar or vector

parameter which does not depend on any domain axis constructs, may have units (such as a reference pressure value) or




may be a descriptive string (such as the projection name "mercator"), or it can be a domain ancillary construct containing data which depends on at least one domain axis construct (such as spatially varying orography data).

For $y$–$x$ coordinates the coordinate conversion is either a map projection, which converts between Cartesian coordinates and spherical or ellipsoidal coordinates on the vertical datum, or a conversion between different spherical coordinate systems
(as in the case of rotated pole coordinates). In the case of $z$ coordinates the conversion is between a coordinate construct with parametrised values (such as ocean sigma coordinates) and a coordinate construct with dimensional values (such as depths), again with respect to the vertical datum.

In some cases the datum is not required as it is already described by the dimension and auxiliary coordinate constructs. This is the case in CF for the two-dimensional geographical latitude–longitude coordinate system based upon a spherical Earth, which
is assumed to have a datum at 0 degrees east, 0 degrees north. Similarly, the coordinate conversion is not required if no other coordinate systems are described. Some parts of the coordinate reference construct may not be relevant to a given coordinate construct which it contains. The relevant parts are determined by an application using the coordinate reference construct. For example, for a coordinate reference construct which contained coordinate constructs for $y$–$x$ projection and latitude and longitude coordinates, a datum comprising a reference ellipsoid would apply to all of them, but projection parameters would
only apply to the projection coordinates.

In CF-netCDF, coordinate system information that is not found in coordinate or auxiliary coordinate variables is stored in a grid mapping variable or the formula_terms attribute of a coordinate variable, for horizontal or vertical coordinate variables respectively. Although these two cases are arranged differently in CF-netCDF, each one contains, sometimes implicitly, a datum or a coordinate conversion formula (or both) and so may be mapped to a coordinate reference construct. A grid mapping name or
the standard name of a parametric vertical coordinate corresponds to a string-valued scalar parameter of a coordinate conversion formula. A grid mapping parameter which has more than one value (as is possible with the "standard parallel" attribute) corresponds to a vector parameter of a coordinate conversion formula. A data variable referenced by a formula_terms attribute corresponds to term of a coordinate conversion formula—either a domain ancillary construct or, if it is zero-dimensional, a scalar parameter.

## 4.5 Domain ancillary construct

A domain ancillary construct (figure 11) provides information which is needed for computing the location of cells in an alternative coordinate system. It is the value of a term of a coordinate conversion formula that contains a data array which depends on one or more of the domain axes.

It also contains an optional array of cell bounds recording the extents of each cell (only applicable if the array contains
coordinate data), and properties to describe the data (in the same sense as for the field construct). An array of cell bounds spans the same domain axes as the data array, with the addition of an extra dimension whose size is that of the number of vertices of each cell.



In general, CF-netCDF variables named by the formula_terms attribute of a CF-netCDF coordinate variable correspond to domain ancillary constructs. These CF-netCDF variables may be coordinate, scalar coordinate or auxiliary coordinate variables; or non-scalar data variables. A scalar data variable (which does not depend on any CF-netCDF dimensions) that is named as a formula term corresponds to a coordinate conversion parameter, rather than a domain ancillary construct. For example, in a coordinate conversion for converting between ocean sigma and height coordinate systems the value of the "depth" term for horizontally varying distance from ocean datum to sea floor would correspond to a domain ancillary construct. In the case of a named term being a type of coordinate variable then that variable will correspond to an independent domain ancillary construct in addition to the coordinate construct.

### 4.6 Cell measure construct

A cell measure (figure 9) construct provides information that is needed about the size or shape of the cells and that depends on a subset of the domain axis constructs. Cell measure constructs have to be used when the size or shape of the cells cannot be deduced from the dimension or auxiliary coordinate constructs without special knowledge that a generic application cannot be expected to have.

The cell measure construct consists of a numeric array of the metric data which spans a subset of the domain axis constructs, and properties to describe the data (in the same sense as for the field construct). The properties must contain a "measure" property, which indicates which metric of the space it supplies e.g. cell horizontal areas, and a units property consistent with the measure property e.g. $m^2$. It is assumed that the metric does not depend on axes of the domain which are not spanned by the array, along which the values are implicitly propagated. CF-netCDF cell measure variables correspond to cell measure constructs.

### 4.7 Field ancillary constructs

The field ancillary construct (figure 9) provides metadata which is distributed over the same sampling domain as the field itself. For example, if a data variable holds a variable retrieved from a satellite instrument, a related ancillary data variable might provide the uncertainty estimates for those retrievals (varying over the same spatiotemporal domain).

The field ancillary construct consists of an array of the ancillary data, which spans a subset of the domain axis constructs, and properties to describe the data (in the same sense as for the field construct). It is assumed that the data does not depend on axes of the domain which are not spanned by the array, along which the values are implicitly propagated. CF-netCDF ancillary data variables correspond to field ancillary constructs. Note that a field ancillary construct is constrained by the domain definition of the parent field construct, but does not contribute to the domain's definition, unlike, say, an auxiliary coordinate construct or domain ancillary construct.





## 4.8 Cell method construct

The cell method constructs (figure 9) describe how the cell values represent the variation of the physical quantity within its cells—the structure of the data at a higher resolution. A single cell method construct consists of a set of axes (see below), a "method" property which describes how a value of the field construct's data array describes the variation of the quantity within a cell over those axes (e.g. a value might represent the cell area average) and properties serving to indicate more precisely how the method was applied (e.g. recording the spacing of the original data, or the fact the method was applied only over El Niño years).

The field construct may contain an ordered sequence of cell method constructs describing multiple processes which have been applied to the data, e.g. a temporal maximum of the areal mean has two components—a mean and a maximum, each acting over different sets of axes. It is an ordered sequence because the methods specified are not necessarily commutative. There are properties to indicate climatological time processing, e.g. multiannual means of monthly maxima, in which case multiple cell method constructs need to be considered together to define a special interpretation of boundary coordinate array values. The cell_methods attribute of a CF-netCDF data variable corresponds to one or more cell method constructs.

The axes over which a cell method applies are either a subset of the domain axis constructs or a collection of strings which identify axes that are not part of the domain. The latter case is particularly useful when the coordinate range for an axis can not be precisely defined, making it impossible to define a domain axis construct. For example, a climatological time mean might be based on data which is not available over the same time periods at every horizontal location—useful information can still be conveyed by recording the fact the data has been temporally averaged without specifying the range of times. The strings which identify such axes are well defined in that they must be standard names (e.g. time, longitude) or the special string "area", indicating a combination of horizontal axes.

## 5 Relationship to other data models

A data model does not exist on its own, and those exploiting it will need to interpret it in the context of other data models with which they already work, whether they are implicit or explicit (section 1.1). Often a clear, unambiguous and universally agreed (or even agreeable) mapping between data models is not possible. Nativi et al. (2008), who establish a mapping between the Unidata Common Data Model and relevant international standards, discuss many of the relevant issues. Here we confine ourselves to drawing some parallels between our explicit CF data model and selected other data modelling activities. We begin with the ISO 19123 coverage model, which provides an abstract view of the problem arena.

Readers who are not familiar with other data models may wish to omit this section on a first reading, as it is not required to understand the CF conventions, the CF data model, nor the software implementation of section 6.



## 5.1 The ISO 19123 Coverage Model

ISO 19123 (International Standards Organisation, 2007) provides a language and conceptual schema for describing "coverages", that is, for datasets which assign data values to specified data locations (in space and/or time). ISO 19123 builds on a range of other ISO standards for geographical information, collectively known as the ISO 191xx series.

An ISO 191xx coverage may be viewed as a function whose inputs are spatiotemporal positions (the "domain") which are related to outputs comprising values of one or more geographical features (the "range" of the coverage). Thus a coverage is notionally a function over a domain which has a range of values. Within ISO 19123 two types of coverage are defined: discrete and continuous. The former is the most relevant here (see figure 12a), having a domain which consists of a set of "Domain Objects", themselves described by spatial objects and temporal primitives. In such a coverage, all the objects must

share the same coordinate reference system. Coordinate reference systems themselves can be compound, but the simplest single coordinate reference system effectively consists of a datum and a set of coordinates (which might include one or more parametric coordinates) defining a coordinate system. Although the language of much of the ISO 19123 specification is cast in terms of simple geospatial coordinates, these abstract coordinate systems are in fact fully general—although many ISO-compliant implementations restrict them to geospatial coordinates, and cannot reflect the full generality of CF coordinates

such as wavelengths, ensembles, etc. UML diagrams (appendix A) for the full ISO 19123 model are given in figure 12.

For point data a discrete coverage is nearly identical to a CF field construct ($V(d)$, as described in section 3). However, a CF field construct explicitly describes a single variable on a domain, whilst a discrete coverage allows for multiple variables on a shared domain. (In ISO 19123, such variables are termed "features", not to be confused with "sampling features" discussed below.) It is also worth noting that many, but not all, netCDF files containing CF data may well have CF data with a shared

implied spatiotemporal domain, and such files may well map rather directly onto the coverage model, but as we have seen, this is not always the case, not least because multiple domain descriptions may appear in a given file.

Discrete coverages themselves can be further specialised into a set of more specialised coverages: sampling the domain with sets of points, a grid of points, or sets of curves, surfaces or solids.

Of these the most important (in terms of our CF data model), is the DiscreteGridPointCoverage (figure 12b). This relates

a domain of grid points laid out in a grid to a range of values in GridValueMatrix. The grid can be defined in three different ways: as a set of grid points (with their own coordinates), as a "rectified" grid—a grid utilising a datum and coordinate vectors for which there is an affine transformation between the coordinates and those of an external coordinate reference system—or a referenceable grid which provides parametric relations between a position using a coordinate tuple and a position in a planetary reference frame using a specific coordinate reference system. The values can appear in a grid matrix, which defines how the

sequence of values is laid out against the positioning exposed via the domain grid (in any of the three forms).

There is a clear correspondence between the CF dimension coordinate construct and an ISO coordinate reference system as used in a rectified grid, and between the underlying concepts of ISO parametric coordinate reference systems within a referenceable grid and a CF domain described using CF auxiliary coordinate constructs. This correspondence together supports




a) Coverage basics: domain, range, and coordinates

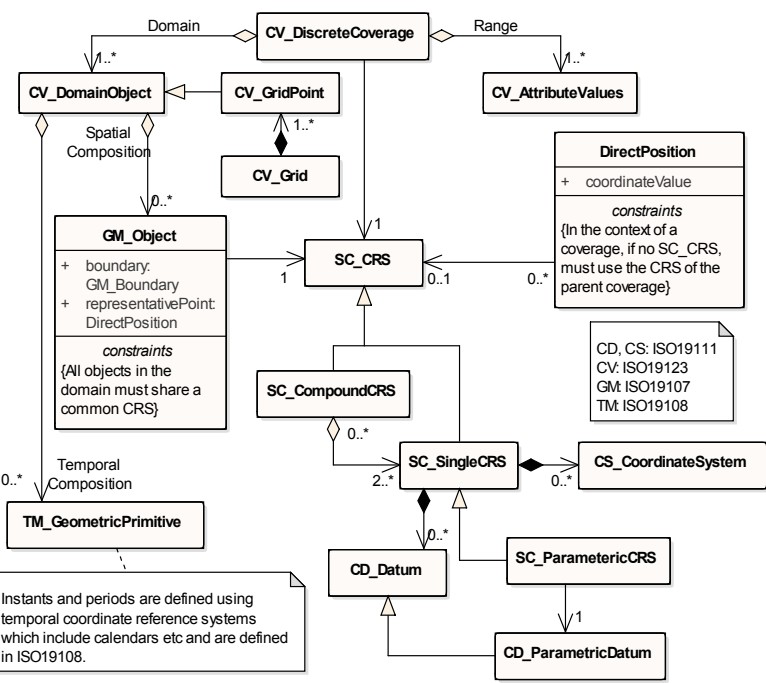

b) Discrete Grid Point Coverages and relation to cells, coordinates, and footprints.

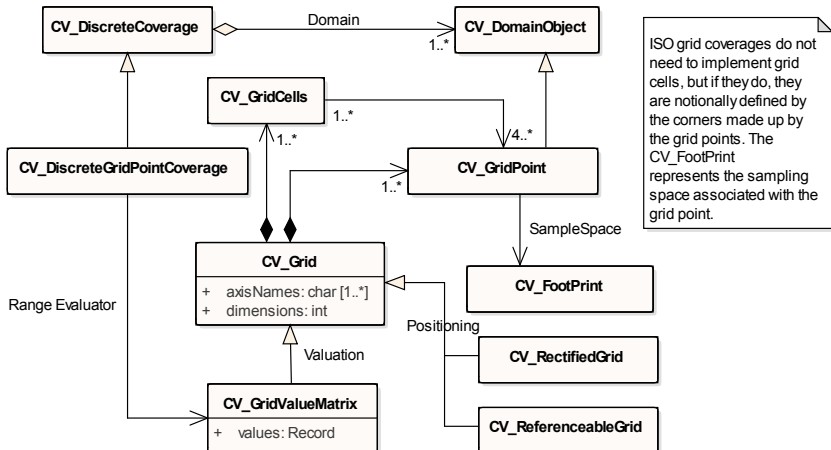

**Figure 12.** Key concepts within the ISO 19123 view of coverages and the associated coordinate systems: a) as applied to a specific coverage: the discrete coverage, which relates a domain of objects to a a set of attribute values; b) an expanded view of the relationship of discrete grid point coverages to grid cells, grid footprints, and three different ways of thinking about the grids (set of coordinates, rectified, and referenceable grids). Note the comment box defining the two-letter labels of each element (e.g. CV means that this elements comes the the ISO 19123 data model). See also figure 9 in Dominico and Nativi (2013). Further relationships and details are discussed in the text.



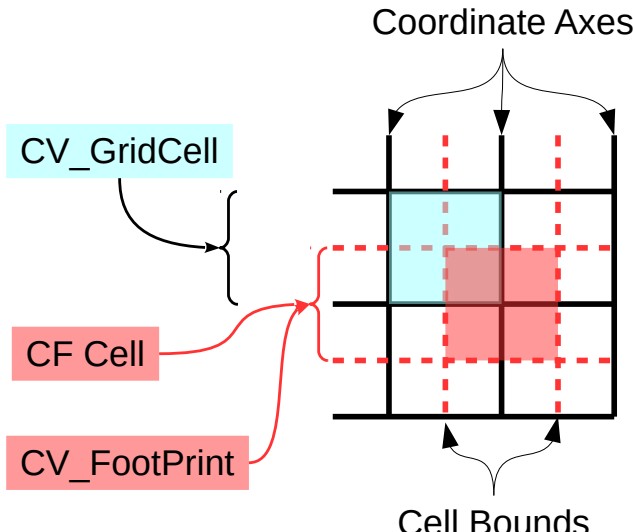

**Figure 13.** The relationship between ISO grid cells and footprints, and CF cells described by cell bounds.

the identification of an ISO grid (which itself carries little information apart from a name and a list of axes) with the abstract

notion of a CF domain described by CF coordinate reference constructs.

Even with an ISO rectified grid, which has the easiest correspondence with CF, there are subtle but important differences in the treatment of coordinates, probably the most important of which is that the CF equivalent of the ISO datum is often held in the standard name of the coordinate construct. For example, a CF coordinate construct with a standard name of height means the coordinate is with reference to the surface, i.e. the bottom of the atmosphere (distinct from other valid vertical coordinates

such as height_above_reference_ellipsoid and height_above_sea_floor). These coordinates are all distinct geophysical quantities, with vertical datums of the surface, the reference ellipsoid and the sea floor respectively; though they all have the same canonical unit of measure (metres) and direction (values increase for locations further above the datum).

Where a more precise specification of the datum may be needed (for example the figure of the reference ellipsoid, or the reference point for a latitude–longitude coordinate system where it is not the default of the intersection of the equator and

the Greenwich meridian) it can be supplied by the coordinate reference construct, not the standard name. This CF separation of grid mapping datum from coordinates adds value because changing the datum does not alter the geophysical nature of the coordinate, and its interpretation. The partitioning is suitable and convenient for many purposes of data analysis, in which coordinate constructs are processed independently, without the need for awareness of a full ISO coordinate reference system (CRS). It arises from the generality of CF, in which spatiotemporal coordinates are used having a wider variety than in GIS;

non-spatiotemporal coordinates are also needed; and the data is often from idealised worlds (such as in climate models).

Whether grid cells are described directly, or implied via referenceable or rectified grids, it is important to note that in the ISO world, the cells lie between the edges laid out by the coordinates, whereas by contrast the notion of a cell in CF—defined by the cell measure construct (section 4.6)—is more directly analogous (figure 13) to the FootPrint in an ISO coverage which





represents the sample space associated with a grid point (figure 12b). There is also a richer set of semantics in CF associated
with the cell method construct (section 4.8).

It might appear that some of the more complex geometries underlying CF fields which are expressed on domains sampled
using the CF DSG features best be mapped onto other specialisations of DiscreteCoverages—this is the approach taken by
Nativi et al. (2008) who use the DiscreteCurveCoverage for mapping ISO coverages onto Unidata Common Data Model
(section 5.3) profile and trajectory data. However, we would assert that the underlying semantics of CF as expressed by the
concept of cells is generalisable to multidimensional sampling inside the DiscreteGridPoint coverage. This is in part why we
see the CF sampling feature types as simply specialisations of the CF field, possibly with a specific storage pattern. This simple
mapping between CF "domains" and ISO DiscreteGridPoint coverage is also the approach taken by the Open Geospatial
Consortium (OGC) CF-netCDF extension standard (section 5.2 below) which describes another set of possible relationships
between CF-netCDF and ISO 19123, albeit without the higher-level CF constructs we introduce here.

ISO 19156 Observations and Measurements (International Standards Organisation, 2011) introduces sampling features with
a range of geometric spatial properties (e.g. sampling along a curve, such as a trajectory). CF discrete sampling geometries can
be mapped onto these sampling features, but it is important to note that ISO 19156 explicitly expects actual observation (or
simulation) values to be sub-sampled. Thus the observations are amenable to recording with more general discrete coverages,
just as we have done here by treating all CF discrete sampling geometries as CF field constructs, optionally labelled with a
feature type property to help understand the intended use of the axes.

## 5.2   The OGC CF-netCDF standard

The Open Geospatial Consortium standard introduced above (Dominico and Nativi, 2013) in the context of complex geometries
presents their own CF data model: the CF-netCDF extension model. Their model differs from ours in three major ways:

1. It is not CF version 1.6 complete (for example, it does not appear to include ancillary data variables),

2. Their model makes some elements of CF mandatory, in order to facilitate the ISO 19123 coverage interoperability which
is their target.

3. It is constructed in order to map as closely as possible onto the ISO 19123 coverage model, but without being faithful
to CF, so for example, it introduces the notion of a CF coordinate system including a notional HorizontalCRS, which
is independent of explicitly identified horizontal and vertical coordinates (their figure 4). By contrast we have only
introduced new concepts as abstractions where they help interpret and use CF itself (again for example, in our case the
domain and abstract coordinate).

The last of these points is of course subjective. We would argue that our approach is the most consistent with a faithful model of
CF, but it is clear that CF itself currently admits a multitude of possible interpretative models as well as a multitude of correct
(if limited and/or constrained) implementations such as this one.





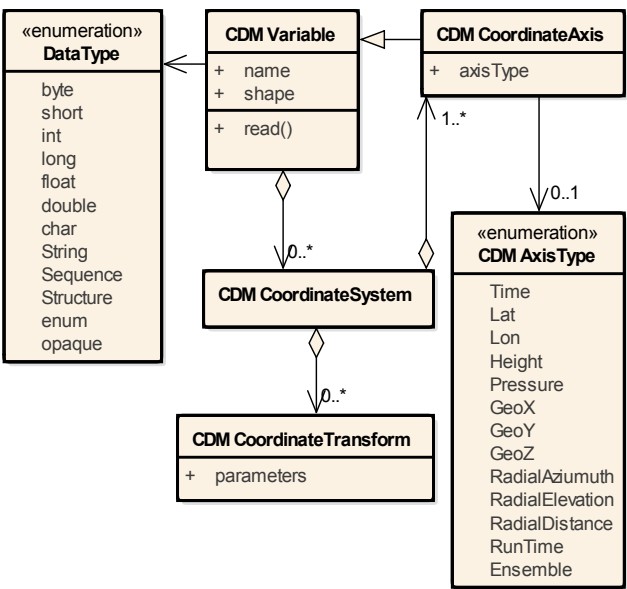

**Figure 14.** Key characteristics of the Unidata Common Data Model. There is a wider variety of fundamental data types than is supported by the netCDF classic data model; and the coordinate system includes the option of coordinate axes of specific types for use in the feature types, which limits the flexibility of the CDM data model.

## 5.3 The Unidata Common Data Model

The Unidata Common Data Model (CDM, Unidata (2014a)) is an abstract data model for scientific datasets that is a superset of the netCDF classic and enhanced data models (section 2). In addition to netCDF, and similarly to the CF data model presented here, it consists of multiple layers:

- a data access layer, which handles data reading and writing, and merges the netCDF enhanced, OPeNDAP (Open-source Project for a Network Data Access Protocol, https://www.opendap.org) and HDF (Hierarchical Data Format, https://www.hdfgroup.org) data models to create a common application programming interface (API).

- a coordinate system layer, which handles the coordinates of data arrays,

- a feature type layer, which handles similar notions to those we express with CF field constructs and the CF sampling feature types, and

- a mature Java based implementation which reads, manipulates, and writes, the CDM sampling features.

The CDM data access layer has broader scope than ours (being about more than just netCDF). If we consider that most of the CF standard as expressed in our data model is about handling coordinates, cells, and domains, then our CF data model corresponds to CDM coordinate system layer, with the various CF feature types of our CF data model field constructs corre-





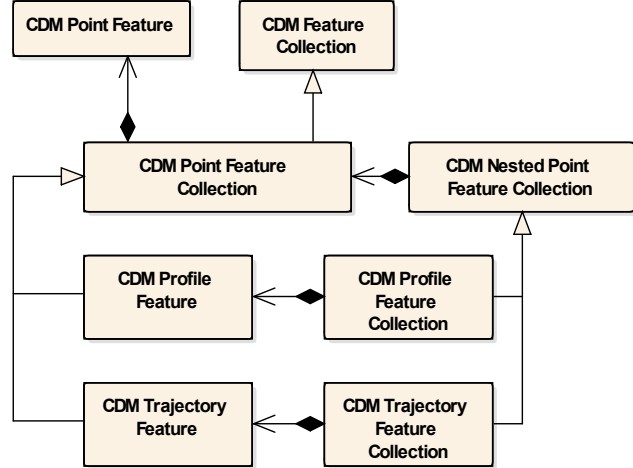

**Figure 15.** The relationship between features, feature types, and feature collections in the Unidata Common Data Model. Only the simplest feature types are shown; more complicated features are built from these basic elements (Unidata, 2014b).

sponding to the CDM feature type layer. The cf-python software (section 6) corresponds to the CDM Java implementation, but the CF data model is intended to be useful outside the context of our cf-python application.

The CDM data access layer handles more data types (figure 14), in particular structures and sequences which allow more complex data types and the iteration through data of (a priori) unknown length. These data types are not yet required by CF-netCDF (section 2), so we do not considered them in our CF data model.

Within the coordinate system layer there is much closer correspondence between the CDM and our CF data model. In particular,

- a CF dimension or auxiliary coordinate construct maps to a CDM CoordinateAxis.

- the datum and coordinate conversion components of a CF coordinate reference construct are components of a CDM CoordinateTransform.

- a CF coordinate reference construct maps to a CDM CoordinateSystem.

The last of these has one exception: a CDM CoordinateSystem must contain at least one CDM CoordinateAxis whereas CF dimension and auxiliary coordinate constructs are optional in a CF coordinate reference construct. In other words, the CF data model can record a coordinate system datum in the absence of coordinate values. This is useful when the CF field construct properties, rather than its coordinates, define the extent of the data array.

A CDM CoordinateAxis may be sub-typed into axes which can be specifically exploited by the sampling feature types in the CDM feature type layer, where there are significant differences from the CF data model. The CDM feature type implementation is discussed in Unidata (2014b); CDM implements feature collections as collections of point features in such a way that a profile for example, is a collection of point features along a vertical line, and so a profile collection is a nested point feature collection.





Similarly, a trajectory feature is a collection of points along a path through time and space, with collections which are nested point feature collections (figure 15). Other CDM feature types are built from this base. The CDM data model exposes these

concepts directly. By contrast, the CF data model does not expose any of these ideas directly, with the interpretation left entirely to software implementations: the CF data model simply exposes the appropriate coordinates and their interpretation is either inferable from the nature of those coordinates, or is made explicit via the feature type property, which is effectively a constraint with a label (section 4.1).

## 6   cf-python: A data model implementation

A key use of our data model is to enable the creation of wholly CF-compliant software, i.e. software that can represent and manipulate any CF-compliant dataset. Such software corresponds to one of the application boxes in figure 1. Cf-python is a data analysis software library written for the Python programming language that implements the CF data model presented here for its internal data structures and so is able to process any CF-compliant dataset. It is, however, not strict about CF-compliance so that partially conformant datasets may be ingested and their deficiencies corrected with the application programming interface

(API). Cf-python is open-source software and is free to download and install from the Python package index at https://pypi. python.org/pypi/cf-python.

Cf-python implements the CF data model constructs and their relationships exactly as shown in figures 9–11, but we refer to the cf-python realisations of CF data model constructs as "objects" in order to distinguish between abstract concepts and their instantiated counterparts. Cf-python (version 2.0) can read field objects from netCDF files or create new field objects,

manipulate field objects in memory and write field objects to netCDF files. Once a field object exists a range of operations are possible, including to:

- create, delete and modify a field object's data and metadata,

- select and subspace field objects according to their metadata,

- perform arithmetic, comparison and other mathematical operations involving field objects,

- collapse axes by statistical operations,

- perform operations with date-time data,

- regrid fields to new domains using the Earth System Modeling Framework high performance software infrastructure (Ryan O'Kuinghttons et al., 2016),

- visualise field objects by interfacing with the cf-plot Python package, which is also open-source and freely available at
https://pypi.python.org/pypi/cf-plot.

All of these operations are "metadata-aware", which means that parameters needed for an operation need not be fully specified by the user, provided that field objects have sufficient metadata to infer the parameters unambiguously. This is greatly



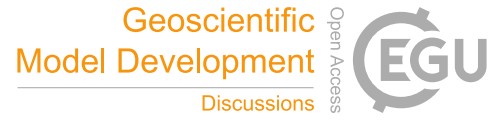

```
>>> import cf
>>> f = cf.read('example_file.nc')
>>> f
[<CF Field: air_temperature(atmosphere_sigma_coordinate(20), projection_y_coordinate(110), projection_x_coordinate(106)) K>,
 <CF Field: atmosphere_mass_content_of_water_vapor(projection_y_coordinate(110), projection_x_coordinate(106)) kg m-2>]
```

**Figure 16.** Reading a file using cf-python.

facilitated by having a data model, because all standardised metadata are stored in a fully defined manner and so the required parameters may be inferred unambiguously. In practice a field object's metadata may be incomplete, in which case the user should use the cf-python API to supplement the metadata. For example, the cf-python command `h=f.regrids(g)` will create a new field object `h` which has the data from field object `f` regridded to the latitude–longitude plane of the domain of field object `g`. If the domains of `f` and `g` are not sufficiently described for this operation then an error will be raised that states which information is missing. The full API documentation is available as part of the cf-python installation, as well as via the Python package index.

How cf-python implements the CF data model may be seen by using the library to read the CF-netCDF file described in figure 3 and inspecting the field objects that it creates. The code listed in figure 16 demonstrates importing the library and reading this netCDF file within the interactive Python shell, in which a command is preceded by the >>> prompt and followed by any printed output. After importing the cf-python library (`import cf`), the netCDF file `example_file.nc` described in figure 3 is read into the variable `f`, which is a list of the file's two field objects—one containing air temperature data and the other vertical integral of atmospheric water vapour data. See section 6.1 for a discussion on why only two field objects were created from the seventeen netCDF variables in the file. The one-line description of each field object shows the size and physical nature of the data array dimensions and the units of the data array values. In figure 17, the first of the two field objects is selected and labelled `t`, and a more detailed look at its metadata is generated with the `dump` function. This exposes the constructs of the data model which have been instantiated for this field object and shows the field object's properties (standard_name and source instantiated from lines 54 and 74 of figure 3). The output describes

- one field object

- four domain axis objects and their sizes, including one which is implied by the "time" CF-netCDF scalar coordinate variable.

- one cell method object indicating that each data array value is a time average constructed from daily samples,

- one field ancillary object describing the uncertainty of the data array values,

- four dimension coordinate objects, each one spanning a unique domain axis object,

- two multidimensional auxiliary coordinate objects for true latitude and true longitude coordinates (as required by the CF conventions when the horizontal dimension coordinates are not canonical geographical latitudes and longitudes),

- three domain ancillary objects utilised by the coordinate reference objects,



```
>>> t = f[0]
>>> t.dump()
---------------------
Field: air_temperature
---------------------
source = 'climate model'
standard_name = 'air_temperature'

Domain Axis: time(1)
Domain Axis: atmosphere_sigma_coordinate(20)
Domain Axis: projection_y_coordinate(110)
Domain Axis: projection_x_coordinate(106)

Data(atmosphere_sigma_coordinate(20), projection_y_coordinate(110), projection_x_coordinate(106)) = [[[276.66, ..., 304.78]]] K

Cell Method: time: mean (interval: 1 day)

Field Ancillary: air_temperature standard_error
    standard_name = 'air_temperature standard_error'
    Data(atmosphere_sigma_coordinate(20), projection_y_coordinate(110), projection_x_coordinate(106)) = [[[1.88, ..., -0.15]]] K

Dimension Coordinate: atmosphere_sigma_coordinate
    positive = 'down'
    standard_name = 'atmosphere_sigma_coordinate'
    Data(atmosphere_sigma_coordinate(20)) = [0.992, ..., 0.003]
    Bounds(atmosphere_sigma_coordinate(20), 2) = [[1.0, ..., 0.0]]

Dimension Coordinate: projection_y_coordinate
    standard_name = 'projection_y_coordinate'
    Data(projection_y_coordinate(110)) = [0.0, ..., 109.0] km
    Bounds(projection_y_coordinate(110), 2) = [[-0.5, ..., 109.5]] km

Dimension Coordinate: projection_x_coordinate
    standard_name = 'projection_x_coordinate'
    Data(projection_x_coordinate(106)) = [0.0, ..., 105.0] km
    Bounds(projection_x_coordinate(106), 2) = [[-0.5, ..., 105.5]] km

Dimension Coordinate: time
    standard_name = 'time'
    Data(time(1)) = [2017-07-01T00:00:00Z] gregorian
    Bounds(time(1), 2) = [[2017-01-01T00:00:00Z, 2018-01-01T00:00:00Z]] gregorian

Auxiliary Coordinate: latitude
    standard_name = 'latitude'
    Data(projection_y_coordinate(110), projection_x_coordinate(106)) = [[75.32, ..., 22.89]] degrees_north

Auxiliary Coordinate: longitude
    standard_name = 'longitude'
    Data(projection_y_coordinate(110), projection_x_coordinate(106)) = [[45.98, ..., 73.57]] degrees_east

Domain Ancillary: atmosphere_sigma_coordinate
    standard_name = 'atmosphere_sigma_coordinate'
    Data(atmosphere_sigma_coordinate(20)) = [0.992, ..., 0.003]
    Bounds(atmosphere_sigma_coordinate(20), 2) = [[1.0, ..., 0.0]]

Domain Ancillary: surface_air_pressure
    standard_name = 'surface_air_pressure'
    Data(projection_y_coordinate(110), projection_x_coordinate(106)) = [[100399.82, ..., 99241.16]] Pa

Domain Ancillary: air_pressure
    standard_name = 'air_pressure'
    Data(projection_y_coordinate(110), projection_x_coordinate(106)) = [[500.46, ..., 498.19]] Pa

Coordinate Reference: atmosphere_sigma_coordinate
    standard_name = atmosphere_sigma_coordinate
    ps = Domain Ancillary: surface_air_pressure
    ptop = Domain Ancillary: air_pressure
    sigma = Domain Ancillary: atmosphere_sigma_coordinate
    Coordinate = Dimension Coordinate: atmosphere_sigma_coordinate

Coordinate Reference: lambert_conformal_conic
    grid_mapping_name = lambert_conformal_conic
    latitude_of_projection_origin = 25.0
    longitude_of_central_meridian = 265.0
    standard_parallel = 25.0
    Coordinate = Auxiliary Coordinate: latitude
    Coordinate = Auxiliary Coordinate: longitude
    Coordinate = Dimension Coordinate: projection_y_coordinate
    Coordinate = Dimension Coordinate: projection_x_coordinate

Cell Measure: area
    standard_name = 'area'
    Data(projection_y_coordinate(110), projection_x_coordinate(106)) = [[2456198746.45, ..., 2500013469.6]] m2
```

**Figure 17.** A detailed inspection of a field object's metadata.



```
>>> t.constructs()
[<CF Field: air_temperature(atmosphere_sigma_coordinate(20), projection_y_coordinate(110), projection_x_coordinate(106)) K>,
 <CF DomainAxis: 106>,
 <CF DomainAxis: 1>,
 <CF DomainAxis: 20>,
 <CF DomainAxis: 110>,
 <CF CellMethod: dim3: mean (interval: 1 day)>,
 <CF FieldAncillary: air_temperature standard_error(20, 110, 106) K>,
 <CF DimensionCoordinate: projection_x_coordinate(106) km>,
 <CF DimensionCoordinate: time(1) gregorian>,
 <CF DimensionCoordinate: atmosphere_sigma_coordinate(20) 1>,
 <CF DimensionCoordinate: projection_y_coordinate(110) km>,
 <CF AuxiliaryCoordinate: latitude(110, 106) degrees_north>,
 <CF AuxiliaryCoordinate: longitude(110, 106) degrees_east>,
 <CF CoordinateReference: lambert_conformal_conic>,
 <CF CoordinateReference: atmosphere_sigma_coordinate>,
 <CF DomainAncillary: atmosphere_sigma_coordinate(20) 1>,
 <CF DomainAncillary: surface_air_pressure(110, 106) Pa>,
 <CF DomainAncillary: air_pressure(110, 106) Pa>,
 <CF CellMeasure: area(110, 106) m2>]
```

**Figure 18.** The cf-python class instances which correspond to CF data model constructs.

– two coordinate reference objects: a vertical, atmosphere sigma coordinate system which references the three domain
     ancillary objects and the vertical dimension coordinate object; and a horizontal, Lambert conformal conic coordinate
     system which references the horizontal auxiliary and dimension coordinate objects,

     – one cell measure object containing horizontal cell areas.

     The one to one correspondence between the CF data model and cf-python's interpretation of CF may also be demonstrated
by inspecting the objects from which field object t is composed. In figure 18 the `constructs` function is used to return all
     of these objects, each having a similar name in camel case to its CF data model counterpart (e.g. a domain axis construct is
     represented by a DomainAxis object). This output demonstrates that it is only the field object which stores information on the
     whole domain. For example, the latitude auxiliary coordinate object has a two-dimensional data array shape of $110 \times 106$, but
     does not record what these dimensions physically represent (figure 10). The cell method object does, however, store references
to the domain axes to which it applies (section 4.8)—the "dim3" in this example refers to the size one domain axis object,
     which is identified by the field object alone as being a time axis by virtue of the "time" dimension coordinate object associated
     with it.

     Whilst field object t contains at least one instance of every type of data model construct, it is more common for field objects
     to contain a subset of the possible data constructs. Figure 19 shows the detailed description of two other cf-python field objects.
The first of these (p) is of medium complexity and contains only domain axis, cell method and dimension coordinate objects.
     In this case, the cell method and time dimension coordinate objects collectively state that the data are 30 year averages of
     monthly minima. The second of the field objects (q) is minimally complex and contains no other data model constructs, yet is
     still CF-compliant. In this case, the data array is scalar and there are no coordinates, so domain axes are not necessary.



```
>>> q.dump()
-----------------------
Field: specific_humidity
-----------------------
long_name = 'specific humidity at atmosphere lower boundary'
standard_name = 'surface_specific_humidity'

Domain Axis: time(12)
Domain Axis: latitude(64)
Domain Axis: longitude(128)

Data(time(12), latitude(64), longitude(128)) = [[[0.006348, ..., 0.098766]]]

Cell Method: time: minimum within years
Cell Method: time: mean over years

Dimension Coordinate: time
    axis = 'T'
    standard_name = 'time'
    Data(time(12)) = [1960-12-16T12:00:00Z, ..., 1961-11-16T00:00:00Z] noleap
    Bounds(time(12), 2) = [[1960-12-01T00:00:00Z, ..., 1990-12-01T00:00:00Z]] noleap

Dimension Coordinate: latitude
    axis = 'Y'
    standard_name = 'latitude'
    Data(latitude(64)) = [-87.8638, ..., 87.8638] degrees_north
    Bounds(latitude(64), 2) = [[-90.0, ..., 90.0]] degrees_north

Dimension Coordinate: longitude
    axis = 'X'
    standard_name = 'longitude'
    Data(longitude(128)) = [0.0, ..., 357.1875] degrees_east
    Bounds(longitude(128), 2) = [[-1.40625, ..., 358.59375]] degrees_east

>>> p.dump()
-----------------------
Field: precipitation_flux
-----------------------
standard_name = 'precipitation_flux'

Data() = 1.45 kg m-2 s-1
```

**Figure 19.** Examples of cf-python field objects of medium and minimal complexity.

### 6.1 Interpreting CF-netCDF files

Any variable in a CF-netCDF file can always be viewed as a data variable in addition to any metadata role it may have, simply by choosing to ignore any other variables that may reference it. For example, a variable that is named by the "coordinates" attribute of a data variable is always an auxiliary coordinate variable (section 3.3), but this metadata status is conferred solely by the "coordinates" attribute, so by ignoring it variable also becomes a data variable.

When a CF-netCDF file is read a decision must be taken as to which variables are the data variables. By default, cf-python
assumes that only unreferenced variables are data variables that instantiate field objects (variables "temp" and "total_wv" in figure 3). It is possible, however to override this default behaviour so that some or all referenced variables instantiate field objects in addition to instantiating other, metadata objects. For example, the variable "PS" in figure 3 will always create a domain ancillary object but may also, if requested, create an independent field object for surface air pressure.



## 7  Summary and conclusions

In this paper we have presented a formal data model for the CF conventions, identifying the fundamental elements of CF and showing how they relate to each other. We have described the CF conventions in terms of their relationship to the physical world (real or simulated) and in terms of their netCDF encoding, and these steps lead to our identifying the elements which contribute to the CF data model. The CF conventions themselves have been influenced by their netCDF encoding, and therefore the CF data model is indirectly influenced by netCDF, although it aims to be independent of the encoding. We have discussed

the relationships of the CF data model to other data models which address the problem of storing data and metadata, and we have presented a software implementation of the CF data model capable of manipulating any CF-compliant dataset.

It is important to note that the CF data model is a description of what CF is, rather than what it ought to be, either in our opinion or anyone else's. We believe that there is little doubt that a CF data model is of considerable value, and this has been recognised by the CF community, who highlight that a CF data model will aid future developments in the CF conventions and

make it easier to create CF-compliant software (http://cf-trac.llnl.gov/trac/ticket/88). In addition, the existence of a data model can help resolve conflicts in the interpretation of the conventions document. For example, a discussion on the CF mailing list regarding the interpretation of CF-netCDF scalar coordinate variables was resolved with some assistance from a developmental version of the CF data model (http://cf-trac.llnl.gov/trac/ticket/104). There are other discussions on unintended ambiguities that have not been resolved, however, and this is unsatisfactory for those who need to manipulate datasets or create software for

that purpose.

Creating an explicit data model before the CF conventions were written would arguably have been preferable. A data model created a priori increases the likelihood that the problem space (i.e. storing and manipulating data and metadata) is fully spanned and encourages coherent implementations, which could be file storage syntaxes or software codes, the latter being a stated goal of CF. For example, in CF-netCDF horizontal and vertical coordinate reference systems are described with very

different structures—the grid mapping variable and formula_terms attribute respectively—a situation that would likely not have occurred if a comprehensive CF data model already existed. Writing a CF data model a posteriori clearly can not bring about all of these benefits, as the coverage of the problem space and file storage syntax are givens, but it can still be of use to software implementations and future developments in the conventions.

We believe that the data model proposed here is a complete and correct description of CF, because we have yet to find a case

for which our implementation in the cf-python library fails to represent or misrepresents a CF-compliant dataset. Moreover, the development of cf-python proves that is possible to implement our CF data model. We consider that our CF data model is simpler and more flexible than other such models, because it defines a small number of general constructs rather than many specialised ones. While the latter approach is closer to an object-orientated software implementation, our aim is to describe CF in a way which is independent of any software.

Version 1.7 of the CF conventions is in preparation and it is our intention is to ensure that the CF data model is updated along with future versions of the CF conventions. In later versions, CF will need to address the representation of data arrays on un-structured domains, such as one for a finite element mesh of triangles. Much of the groundwork for describing unstructured do-




main representations has already been done by the UGRID conventions (http://ugrid-conventions.github.io/ugrid-conventions), and this may become integrated into CF. An unstructured data array may not be representable as a hyperrectangle in index space, and it is possible that the data model presented here will need extending when these are formally included in the CF conventions.

Finally, we believe that it would be desirable for the CF data model to be formally incorporated into the CF conventions to act as a reference for interpreting and extending CF. This would also ensure that as the CF conventions evolve the CF data model gets updated, if necessary, by the CF community.

## 8  Code availability

The Python source code for cf-python is open-source and freely available from its on-line repository (https://bitbucket.org/cfpython/cf-python). The source code is also downloaded when cf-python is installed from the Python package index (https://pypi.python.org/pypi/cf-python).

## Appendix A:  A UML primer

Throughout this paper we rely on the Unified Modelling Language (UML) to construct diagrams that define the key relationships of the entities described in CF-netCDF files and in our data model. These diagrams show relationships between "classes" like those used in an object-orientated programming language or like data types in Fortran. The relationship of an "instance of a class" to its class is like that of a particular variable to its data type. A class is like a species of animal, and an instance of a class is like an individual animal. Classes can be included in other classes, just as components are included in the definitions of derived data types in Fortran, and organs comprise the body of an animal.

For reference in interpreting our UML diagrams, we describe the subset of UML used here. As depicted in table 1, arrows and symbols are used to show different types of relationship between classes. Some relationships include a "cardinality" which indicates the number of instances of one class that may be associated with an instance of another. If there is a number (say, $n$) present at class Y where there is an arrow from class X to class Y, it indicates that there must be exactly $n$ instances of class Y associated with class X. These cardinalities can also be associated with ranges. For example, $0..1$ means an zero or one instances of class Y may be associated with class X and $0..*$ means any number of instances of class Y may be associated with class X. All of these relationships, along with techniques used to add further information to classes and associations, are shown in the worked example of figure A1.

The UML diagram elements relating to netCDF (section 2), the CF-netCDF encoding and the CF data model (section 4) are coloured yellow, blue and green respectively. In addition, an element from a data model that is not the main focus of the diagram has its name prefixed with an identifier for its model—"NC" for netCDF and "CN" for CF-netCDF. For example, in figure 8, which is focused on the CF-netCDF conventions, the yellow "NC::Dimension" element is the same as the "Dimension" element from figure 2, the main diagram for the netCDF data model.



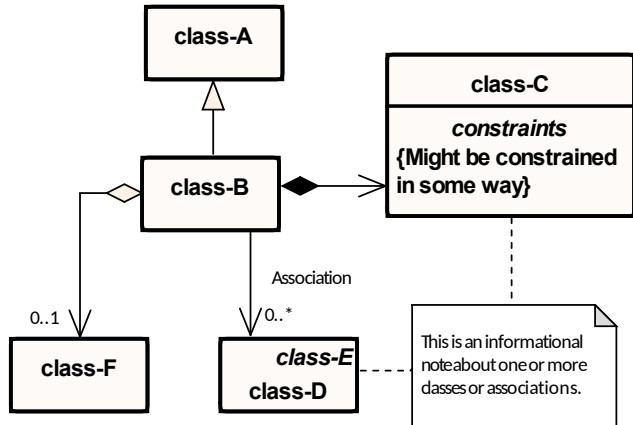

**Figure A1.** A worked example demonstrating the subset of UML used in this paper (see table 1 for definitions of the class associations). Class-B is a subclass of class-A, and class-D is subclass of class-E. An instance of class-B includes one instance of class-C (that can not exist independently) and may include zero or one instances of class-F (that can exist independently). An instance of class-B is related to any number of instances of class-D (with the relationship being described by the label "Association"). An instance of class-C is constrained to exhibit some behaviour. There is a general comment concerning class-D and class-C.

**Appendix B: Optimising dataset storage and creation in CF-netCDF**

An important goal of the CF conventions is that datasets should be efficient to create, store and subsequently read, where efficiency is a measure of the time taken for software to carry out a task or the amount of computer data storage required for a file. The conventions describe various techniques for optimising these requirements. A CF-netCDF file that uses any of the optimisation techniques can always be recast without them and still contain exactly the same scientific information, therefore the optimisation mechanisms do not affect the CF data model described in section 4. It is common for a CF-netCDF file to

not require optimisation, but in the cases where it may be applied, an unoptimised file could suffer by being less readable by humans, consuming more storage or being slower to create.

The example file shown in figure 3 does not include any of these techniques, but many examples may be found in the CF conventions document (Eaton et al., 2011).

**B1  Packing**

Storage space in netCDF files may be reduced by a packing, i.e. by altering the data in a way that reduces its precision. This is achieved through the simple use of the variable attributes "scale_factor" and "add_offset". After the data values of a variable have been read, they are to be multiplied by the scale_factor, and have add_offset added to them (if both attributes are present, the data are scaled before the offset is added). Unpacked values are assumed to have the same data type as the packing attributes, thus making it possible to store floating point data as small integers, say. For example, the 64-bit floating



point number 283.1578 could be stored as the 32-bit integer 100 alongside a scale_factor of 0.1 and an add_offset attribute of 273.15. The unpacked value in this case is 283.15, demonstrating a loss of precision relative to the original data.

## B2    Compression

Space may be saved in netCDF datasets if unwanted array values, that would otherwise have to be stored as missing data, are removed. Such compression techniques store the data more efficiently and result in no precision loss.

### B2.1    Gathering

Compression by gathering combines axes of a multidimensional array into a new, discrete axis (the "list" dimension) whilst omitting the missing values and thus reducing the number of values that need to be stored. The information needed to uncompress the data is stored in a separate variable (the "list" variable) that contains the indices needed to uncompress the data. A list variable is encoded as a coordinate variable that has a "compress" attribute which names the dimensions that have been 825 compressed. For example, a variable that spans $x$, $y$ and $t$ axes but has missing data values at all points over the ocean could have its $x$ and $y$ dimensions compressed to a new dimension (called $landpoint$, say) whose size is the number of land points. The stored variable would then span only the $landpoint$ and $t$ dimensions.

### B2.2    Ragged array representations

A collection of discrete sampling geometry (DSG) features may be stored using the contiguous or indexed ragged array rep-830 resentation, which minimise the amount of file storage required (section 3.4). In both cases the "instance" dimension that distinguishes between different features is combined with the number of elements of each feature to create a compressed "sample" dimension. The entire collection may then be stored in an array that spans the sample dimension, and the contiguous and ragged array representations provide different techniques for populating this array and uncompressing it to find the values for individual features.

In the contiguous case each feature in the collection occupies a contiguous block, and so can be used only if the size of each feature is known at the time that it is created. It requires a "count" variable that gives the size of each block and is encoded as a netCDF variable with a "sample_dimension" attribute that names the sample dimension.

   For indexed ragged arrays, the values of each feature in the collection are interleaved along the sample dimension. The canonical use case for this representation is the storage of real-time data streams that contain reports from many sources; 840 the data can be written as it arrives. It requires an "index" variable that specifies the feature that each element of the sample dimension belongs to and is encoded as a netCDF variable with an "instance_dimension" attribute that names the instance dimension.

   It is also possible to combine contiguous and indexed ragged array representations, which is useful for cases such as writing real-time data streams that contain vertical profiles from many trajectories, arriving randomly, with the data for each entire 845 profile written all at once.





*Acknowledgements.* We would like to thank Mark Hedley, Antonio Cofiño, Martin Juckes, Alison Pamment and Paulo Ceppi for comments that greatly improved the manuscript. We are also indebted to members of the CF community, whose considerable efforts ensure the continuing success of the CF conventions; in particular those who took part in the data model discussions that took place on the CF mailing list.

The research leading to these results has received funding from the core budget of the U.K. National Centre for Atmospheric Science, the European Research Council, and the European Commission's Seventh Framework programme (from ERC project "Seachange", number 247220; and FW7 project "IS-ENES2", number 312979). Work by Karl E. Taylor was performed under the auspices of the U.S. Department of Energy (USDOE) by Lawrence Livermore National Laboratory under contract DE-AC52-07NA27344 with support from the Regional and Global Climate Modeling Program of the USDOE's Office of Science.



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
