# Peer review of "A data model of the Climate and Forecast metadata conventions (CF-1.6) with a software implementation (cf-python v2.1)"

_Geoscientific Model Development, 2017_

## Short Comment (SC1) · 18 Jul 2017

Dear authors,

In my role as Executive editor of GMD, I would like to bring to your attention our Editorial version 1.1:

http://www.geosci-model-dev.net/8/3487/2015/gmd-8-3487-2015.html

This highlights some requirements of papers published in GMD, which is also available on the GMD website in the 'Manuscript Types' section:

http://www.geoscientific-model-development.net/submission/manuscript_types.html

In particular, please note that for your paper, the following requirements have not been met in the Discussions paper:

[Figure]

- "The main paper must give the model name and version number (or other unique identifier) in the title."

- "If the model development relates to a single model then the model name and the version number must be included in the title of the paper. If the main intention of an article is to make a general (i.e. model independent) statement about the usefulness of a new development, but the usefulness is shown with the help of one specific model, the model name and version number must be stated in the title. The title could have a form such as, "Title outlining amazing generic advance: a case study with Model XXX (version Y)"."

- "All papers must include a section, at the end of the paper, entitled 'Code availability'. Here, either instructions for obtaining the code, or the reasons why the code is not available should be clearly stated. It is preferred for the code to be uploaded as a supplement or to be made available at a data repository with an associated DOI (digital object identifier) for the exact model version described in the paper. Alternatively, for established models, there may be an existing means of accessing the code through a particular system. In this case, there must exist a means of permanently accessing the precise model version described in the paper. In some cases, authors may prefer to put models on their own website, or to act as a point of contact for obtaining the code. Given the impermanence of websites and email addresses, this is not encouraged, and authors should consider improving the availability with a more permanent arrangement. After the paper is accepted the model archive should be updated to include a link to the GMD paper."

Therefore please provide the CF version number in the title (e.g., "A Climate and Forecast data model and implementation for CF 1.6"). Additionally, to ensure permanent access to the exact version the article refers to, please upload the source codes to a permanent archive providing a DOI (e.g. Zenodo).

[Figure]

Yours,

Astrid Kerkweg

---

## Author Comment (AC1) · 21 Jul 2017

Dear Astrid Kerkweg,

Thank you very much for your comments, which we would like address here.

1.  We will change the title from "A CF data model and implementation" to:

    A data model of the Climate and Forecast metadata conventions (CF-1.6) with a software implementation (cf-python v2.0)

2.  I have created a DOI for the cf-python software (doi.org/10.5281/zenodo.832255) and will replace the whole of section 8 (Code availability) with:

    The code of cf-python is open-source and freely downloadable at doi.org/10.5281/zenodo.832255. It is also available from its on-

line repository at https://bitbucket.org/cfpython/cf-python and from the
Python package index at https://pypi.python.org/pypi/cf-python.

I hope that these changes now meet the requirements.

All the best,

David Hassell

---

## Referee Comment (RC1) · V. Balaji (Referee) · 4 Aug 2017

The paper is an in-depth discussion of the CF conventions, which are becoming indispensable for producing climate data conforming to a set of agreed-upon standards. They ensure that software can be built that in principle can process any conforming dataset.

The authors go further and formalize the underlying data model. The data model is an exact and comprehensive formal representation of every possible facet of the CF conventions.

Further, and finally, the paper describes cf-python, a reference implementation of the data model. Software built upon cf-python is guaranteed to be able to express and process any possible CF-conformant dataset.

[Figure]

As far as I can tell, the description of CF is precise and complete; the data model is an utterly faithful replica; and I have no reason to doubt that the software is as well.

I have little hesitation in recommending for publication. The bulk of the content of the paper (Sections 1.1-5.3) requires very little revision. Beautifully written and rigorously edited... no typos or grammar errors I was able to find.

I would however require some revisions to the beginning and end of the text (Sections 1 and 6). I feel the need for a data model isn't sufficiently well motivated; the current problems and anxieties surrounding CF aren't well laid out. Nor is it clear from reading the article how CF may evolve in the future, and what the CF data model will do to guide this evolution. With these changes, I think this paper stands a good chance of becoming a canonical citation for CF, much the same as the Nativi et al paper is for the Common Data Model underlying netCDF..

Let me pose a few questions, and let's see if the answers could help guide the kinds of modifications I would suggest.

1. The data model has been written to be independent of the underlying encoding (L74), viz. netCDF. Nonetheless, the data model is written to conform as far as possible to the netCDF classic format (L83). There have been quite some discussions on whether to introduce new features in CF that in fact require a departure from the classic format. People have pointed to the lack of strings in netCDF-classic (see L92 and L198 in this paper), and whether the use of "groups" and other features imported from HDF into netCDF4 would in fact be a good thing. There are strong opinions on either side of this (including mine..) but without adjudicating this issue, do you think the abstract data model is versatile enough to be forward-compatible (as well as backward-compatible, L29)?

2. There are already features emerging that break the dependence on files, treating "atomic datasets" as synonymous with files. This is likely to be even more so in the feature, as technologies move away from filesystems to object-store. Could you comment

on how the data model will enhance, or inhibit, such a migration? Examples include the "external_variables" attribute that allows the referent of a cell_measures attribute to be in a different, unspecified, location. How would the data model, and cf-python, deal with such cross-references?

3. There have been proposed extensions to CF including Gridspec and UGRID for unstructured grids. Is the data model up to the task of representing such features as say, grid mosaics, which may involve more cross-file references? Or are such concerns entirely outsourced to ESMF and not of concern to the CF data model itself?

4. In general, it has become hard to evolve CF. This is not a problem with CF itself, rather with the sociology of standards: changes require a sufficient number of people to pay sufficient attention in a reasonable amount of time. If the CF data model became the canonical representation, and cf-python the reference implementation, one approach would be to make all proposed changes visible in the data model, and perhaps we could also require the proposer to submit a pull request implementing the suggested change in cf-python. This would give the debates greater clarity; but would they make the problem of evolving the conventions even harder?

5. If the conventions don't evolve rapidly enough, usually small communities break away and work on their own "fork" of the conventions (see UGRID). This could become explicit using the data model: in fact this would mean an actual fork of cf-python, which could make it back on the git trunk at a later date, or not at all. Would this be a good outcome?

I believe some discussion of these questions would greatly enhance the paper.

Minor comments below:

L125: missing value and fill value aren't the same thing. It might be convenient to set them to the same value, but I can think of use cases where you would not want that.

L810: more than reducing precision, this reduces to the representation to integer. Thus

dynamic range is limited (packing to 8-bits means the lowest and highest values after the offset have to be within a factor of 256), and values are discretely spaced.

L819: Currently the netCDF4 library has lossless compression based on zlib... others (Dennis et al, https://www.geosci-model-dev.net/9/4381/2016/gmd-9-4381-2016.html) are proposing lossy compression.

L19, L34, L128, L318, L660, L675, L745, L776, L778, L859, L861, L869, L875, L876: convert to hyperlink (use hyperref package) L36, L140: use \equiv instead of = for definitions L126, L169, L172, L174, L179, L189, L195, L274, L299, L306, L327, L328, L334, L335, L337, L341, L342, L370, L371, L375, L377, Fig 11 caption, L570, L811, L841: for CDL entities in the main text use \texttt{} instead of quotes. L673,L872: omit "Ryan".
* * *

---

## Referee Comment (RC2) · B. E. Eaton (Referee) · 17 Aug 2017

The paper contains complete and concise descriptions of the NetCDF data model and the CF metadata conventions. It presents a generalized data model for CF and describes how this data model represents the CF metadata. The data model is compared with three others, and finally an implementation in python is presented.

In what follows the CF metadata conventions are referred to as CF-netCDF, and a dataset which conforms to those conventions as a CF-netCDF dataset.

This paper is very well written and is a significant contribution to the processing of CF-netCDF datasets. The presentation of cf-python as an implementation of the CF data model is on its own standing sufficient for me to recommend publication. However my point of view concerning how the CF data model relates to CF-netCDF differs in several respects from that expressed in the paper. I have made comments below that,

if addressed, I feel would help to clarify this important relationship.

I don't agree with the paper's contention that CF-netCDF lacks an explicit data model and that such a data model is necessary for processing a CF-netCDF dataset (L39, L42). In the introduction of Nativi et al. (2008) CF-netCDF is called out as an example of a "community standard data model". And later in that paper figure 3 presents the CF-netCDF data model. This suggests that claiming the new CF data model provides an explicit data model for CF-netCDF is not focussing on the most important difference between the two. In fact CF-netCDF "identifies the elements of the dataset and their scientific intent, and describes how they are related to one another and to the real or model world from which the data were derived" (L37). The difference is that CF-netCDF does this at the file level, hence the close connection to the elements of the netCDF data model are necessary. On the other hand, the CF data model describes the same features, but does so at a more general level and without referring to the netCDF data model.

L46 states that the CF data model enables CF-netCDF to be presented in a manner that's easier to understand. I think this is valid. But I don't see how "adherence to the data model will ensure the production of CF-compliant datasets". It is a correct mapping of the elements of the CF data model to the elements of the CF-netCDF data model that ensures production of CF-compliant datasets. If an application or library does this successfully, then users of the application or library API will produce CF-compliant datasets.

I feel that Figure 1 is misleading at best. To say that CF-netCDF allows multiple interpretations implies that the data can be interpreted in different ways. This would imply that the elements of CF-netCDF are not well defined. If there are ambiguous elements of CF-netCDF then it is not the responsibility of the CF data model to rectify this. The problems must be corrected in CF-netCDF.

This leads to what I consider to be the most important contribution made by this work.

The cf-python library is presented as an implementation of the CF data model, but in fact it could just as well be considered a reference implementation of CF-netCDF. This to my knowledge has not been previously accomplished, and is an extremely valuable contribution to the community. Having a reference implementation is a powerful tool to help in verifying that the CF-netCDF data model is well defined. It also has great potential to be used as a testbed which could facilitate future development of CF-netCDF.

At L529 readers not familliar with data models are encouraged to move on. I would suggest removing section 5 as it doesn't really add to the description of the CF data model. The Unidata CDM and the OGC CF-netCDF standards are both concerned with mapping parts of CF-netCDF to the relevent ISO standards. Since that is not a goal of the CF data model the comparison of various data model elements in this section seems extraneous.

Similarly I would suggest that appendix B on data compression in CF-netCDF is extraneous information that is easily found in the CF Convention document.

If space is freed up by the above deletions, I would find it more useful to have a summary of the cf-python API. The paper does a nice job of illustrating how the library faithfully represents all features of an existing CF-netCDF file, but doesn't say anything about how the API would be used to create a new CF-netCDF file.

Finally, one issue I've found with cf-python (v2.0.3) is that it encounters an error when attempting to open a CF-netCDF dataset which contains only coordinate variables. The error message is "RuntimeError: No fields found from 1 files". This seems to be related to the CF data model's treatment of the field construct as a container for all other constructs. Having a file contain only domain information is useful in practice, particularly in the context of interpolating fields from one domain to another.

Minor points:

L119: CF uses the term "coordinate variable" exactly as it was defined by the NUG (as

per section 1.2 in the CF convention document). The motivation for the term "auxiliary coordinate variable" was to avoid the need for the unwieldy phrase "coordinate variable in the NUG sense", i.e., auxiliary coordinate variables contain coordinate data, but are not coordinate variables.

L143: There is nothing in CF that says a file can't just contain coordinate variables, i.e., it's possible that N=0 and M>0. So it's not necessary that M<=N.

L149, L154: t is not a coordinate variable of temp. It's a scalar coordinate variable. There is no t dimension.

L175: I think it's misleading to describe auxiliary coordinate variables as providing "additional or alternative" coordinate information in the sense that that description makes the information sound optional. I would say that the most important use of auxiliary coordinate variables is to provide *required* coordinate information, as discussed starting in line 182.

Figure 8: I'm not sure what this diagram adds. L346 states it could be interpreted as a data model for CF, but I don't see how. For example there is no way from the diagram to know that a coordinate variable is a one dimensional array with a dimension name that matches the variable name and contains strictly monotonic data with no missing values. Figure 3. from Nativi et al. (2008) does a better job of expressing the CF-netCDF data model, though it too is incomplete.

L366: Note that the "units" attribute is not optional in most cases.

Figure 9: Note that "Auxiliary" is misspelled in the «construct» box. Also in Figure 10.

---

## Author Response (AR1)

We very much appreciate the comments from the reviewers and the editor. Considering them has produced an improved manuscript. Here we respond in detail to all of the comments, making frequent references to line numbers in the marked-up mansuscript that incorporates the changes resulting from the reviews. This new, marked-up manuscript is given after the individual responses.
* * *
**SC1 from A. Kerkweg**

Please provide the CF version number in the title (e.g., "A Climate and Fore- cast data model and implementation for CF 1.6").

We have changed the title to "A data model of the Climate and Forecast metadata conventions (CF-1.6) with a software implementation (cf-python v2.1)".

Additionally, to ensure permanent access to the exact version the article refers to, please upload the source codes to a permanent archive providing a DOI (e.g. Zenodo)

We have created a DOI for the cf-python software (doi.org/10.5281/zenodo.832255) and have included this in section 9 on code availability (L855-L858).
* * *
**RC1 from V. Balaji**

I feel the need for a data model isn't sufficiently well motivated; the current problems and anxieties surrounding CF aren't well laid out. Nor is it clear from reading the article how CF may evolve in the future, and what the CF data model will do to guide this evolution.

We give the general motivation in Section 1. Briefly, a data model is needed for interpreting the data, helps in understanding the conventions and their relationship to other data formats, and guides the developers of software intended to process the data and of the convention itself. We have redrafted the introduction (section 1) to remove duplication and improve the logical flow (L36-L73) We do not believe that this paper is the right place to address the outstanding issues with CF. Our motivation here is to provide tooling (data model, cf-python) to better understand the current status of CF, although we hope our tooling will enable CF to evolve with more clarity in the future. We share some of the concerns of the reviewer, and address them with a new section "Evolution of a CF data model and cf-python" (L771-L805). In short, CF will continue to to evolve as it does at present, driven by the community's needs, and a CF data model will have to adapt to changes in the conventions. However, a data model could also be useful in guiding future changes.

The data model has been written to be independent of the underlying encoding (L74), viz. netCDF. Nonetheless, the data model is written to conform as far as possible to the netCDF classic format (L83).

It is the CF conventions (not the data model) that must conform to the netCDF3 classic format. The text has been updated to make this distinction clearer (L96).

There have been quite some discussions on whether to introduce new features in CF that in fact require a departure from the classic format. People have pointed to the lack of strings in netCDF-classic (see L92 and L198 in this paper), and whether the use of "groups" and other features imported from HDF into netCDF4 would in fact be a good thing. There are strong opinions on either side of this (including mine..) but without adjudicating this issue, do you think the abstract data model is versatile enough to be forward-compatible (as well as backward-compatible, L29)?

We suggest that this data model is designed to be as flexible as possible and is therefore likely (but not guaranteed) to meet future requirements (new L772-L776).

There are already features emerging that break the dependence on files, treating "atomic datasets" as synonymous with files. This is likely to be even more so in the feature, as technologies move away from filesystems to object-store. Could you comment on how the data model will enhance, or inhibit, such a migration? Examples include the "external_variables" attribute that allows the referent of a cell_measures attribute to be in a different, unspecified, location. How would the data model, and cf-python, deal with such cross-references?

The data model will most likely be neutral with regards these issues, as they are artifacts of the encoding rather than the logical content of a data set, and have included the example being able to store a cell measures variables in a different file (L781-L788).

There have been proposed extensions to CF including Gridspec and UGRID for unstructured grids. Is the data model up to the task of representing such features as say, grid mosaics, which may involve more cross-file references? Or are such concerns entirely outsourced to ESMF and not of concern to the CF data model itself?

Commenting on future proposals is beyond the scope of this paper (which restricts its design to CF-1.6), but the potential suitability of the data model to be able to cope is discussed in the new text at L772-L776.

In general, it has become hard to evolve CF. This is not a with CF itself, rather with the sociology of standards: changes require a sufficient number of people to pay sufficient attention in a reasonable amount of time. If the CF data model became the canonical representation, and cf-python the reference implementation, one approach would be to make all proposed changes visible in the data model, and perhaps we could also require the proposer to submit a pull request implementing the suggested change in cf-python. This would give the debates greater clarity; but would they make the problem of evolving the conventions even harder?

We support the adoption in CF of a canonical data model (L838-L854) with a reference implementation, but feel that putting the responsibility of updating the reference implementation onto an individual proposer of a CF enhancement would be too onerous. This is further described in the new text at L797-L805.

If the conventions don't evolve rapidly enough, usually small communities break away and work on their own "fork" of the conventions (see UGRID). This could become explicit using the data model: in fact this would mean an actual fork of cf-python, which could make it back on the git trunk at a later date, or not at all. Would this be a good outcome?

Yes, it would be good outcome if a reference implementation (cf-python or otherwise) could be forked and updated by interested parties. See the new text at L797-L805.

L125: missing value and fill value aren't the same thing. It might be convenient to set them to the same value, but I can think of use cases where you would not want that.

We have replaced the example of _FillValue with that of missing_data, as a better example of the general concept (L140-L142).

L810: more thaan reducing precision, this reduces to the representation to integer. Thus dynamic range is limited (packing to 8-bits means the lowest and highest values after the offset have to be within a factor of 256), and values are discretely spaced.

Text has been added to appendix B to clarify this (L899-L903).

L819: Currently the netCDF4 library has lossless compression based on zlib... others (Dennis et al, https://www.geosci-model-dev.net/9/4381/2016/gmd-9-4381-2016.html) are proposing lossy compression.

We ackowledge that the CF methods for lossless and lossy compression aren't the only way to do it, and have added text discussing this at L889-L892 (lossless compression) and L895-L896 (lossy compression).

L19, L34, L128, L318, L660, L675, L745, L776, L778, L859, L861, L869, L875, L876: convert to hyperlink (use hyperref package)

Fixed (many lines).

L136, L140: use \equiv instead of = for definitions

Fixed (L152 and L156).

L126, L169, L172, L174, L179, L189, L195, L274, L299, L306, L327, L328, L334, L335, L337, L341, L342, L370, L371, L375, L377, Fig 11 caption, L570, L811, L841: for CDL entities in the main text use \texttt{} instead of quotes.

Fixed (many lines).

L673,L872: omit "Ryan".

Fixed (L701 and L959).
* * *
**RC2 from Brian Eaton**

I don't agree with the paper's contention that CF-netCDF lacks an explicit data model and that such a data model is necessary for processing a CF-netCDF dataset (L39, L42). In the introduction of Nativi et al. (2008) CF-netCDF is called out as an example of a "community standard data model". And later in that paper figure 3 presents the CF-netCDF data model. This suggests that claiming the new CF data model provides an explicit data model for CF-netCDF is not focussing on the most important difference between the two. In fact CF-netCDF "identifies the elements of the dataset and their scientific intent, and describes how they are related to one another and to the real or model world from which the data were derived" (L37). The difference is that CF-netCDF does this at the file level, hence the close connection to the elements of the netCDF data model are necessary. On the other hand, the CF data model describes the same features, but does so at a more general level and without referring to the netCDF data model.

We believe that CF, as currently defined, is not explicit about the structure and interpretation of data, and it is clearly not defined using any data model formalism. We agree that Nativi et al provide "a" data model, but we do not believe it is "the" data model, any more than we believe ours is(L43-L62). Indeed, the first word of our paper title is "A". Neither Nativi's, nor ours, is currently an agreed CF data model, but we assert that the Nativi model is less satisfactory than ours, because it's not complete and not consistent with the CF conventions. In the section about their model, we add a fourth point, that it's tied to the netCDF format (L635). At the end of section 1.1 we make independence of the file format a requirement of the data model (L86-L88). We accept that we did drift into the use of the phrase "the data model" where "a data model" would be better. We have corrected this.

L46 states that the CF data model enables CF-netCDF to be presented in a manner that's easier to understand. I think this is valid. But I don't see how "adherence to the data model will ensure the production of CF-compliant datasets". It is a correct mapping of the elements of the CF data model to the elements of the CF-netCDF data model that ensures production of CF-compliant datasets. If an application or library does this successfully, then users of the application or library API will produce CFcompliant datasets.

A good point. We retract the comment about the production of CF-compliant datasets and have reworded that section (L70-L73).

I feel that Figure 1 is misleading at best. To say that CF-netCDF allows multiple interpretations implies that the data can be interpreted in different ways. This would imply that the elements of CF-netCDF are not well defined. If there are ambiguous elements of CF-netCDF then it is not the responsibility of the CF data model to rectify this. The problems must be corrected in CF-netCDF.

We do not mean to say that the data is unclear in its intention, but that there are some choices to be made about how the data is to be conceptualised.  One example, now resolved by an explicit statement in CF version 1.7, is whether scalar coordinate variables imply the existence of a dimension of the data; this wasn't stated in the convention before because it doesn't affect the data itself, but it may affect the design of software to implement CF.  A more basic example is the idea that the field is a central concept of CF, implying that it's naturally represented in software as a single object.  This is not stated explicitly in the convention, but we state it explicitly in the data model. We have added comments to the text (L43-L73) and the caption to figure 1.

This leads to what I consider to be the most important contribution made by this work.  The cf-python library is presented as an implementation of the CF data model, but in fact it could just as well be considered a reference implementation of CF-netCDF. This to my knowledge has not been previously accomplished, and is an extremely valuable contribution to the community. Having a reference implementation is a powerful tool to help in verifying that the CF-netCDF data model is well defined. It also has great potential to be used as a testbed which could facilitate future development of CF-netCDF.

We are glad that the reviewer sees potential for uses of cf-python. These points are discussed in the new text at L797-L805.

At L529 readers not familiar with data models are encouraged to move on. I would suggest removing section 5 as it doesn't really add to the description of the CF data model. The Unidata CDM and the OGC CF-netCDF standards are both concerned with mapping parts of CF-netCDF to the relevant ISO standards. Since that is not a goal of the CF data model the comparison of various data model elements in this section seems extraneous.

We clearly have sympathy with this view, as can be seen by our encouragement in L557, but would like to argue for this section to remain. Interfacing CF with other international standards is becoming ever more important and we hope that, for some readers, how our CF data model relates to other data models will provide valuable context.

Similarly I would suggest that appendix B on data compression in CF-netCDF is extraneous information that is easily found in the CF Convention document.

Appendix B was included for two reasons: firstly to provide a complete yet brief summary of CF, which doesn't to our knowledge exist elsewhere, and secondly to provide evidence for our claim that these mechanisms do not affect our CF data model (L125-L127). However, we leave this to the discretion of the editor.

If space is freed up by the above deletions, I would find it more useful to have a summary of the cfpython API. The paper does a nice job of illustrating how the library faithfully represents all features of an existing CF-netCDF file, but doesn't say anything about how the API would be used to create a new CF-netCDF file.

We have a separate paper planned to further describe the practical use of cf-python for reading, creating, manipulating and writing fields, so would prefer to keep these aspects of cf-python to a brief summary in this paper, as they do not directly impact on our understanding of the CF data model.

Finally, one issue I've found with cf-python (v2.0.3) is that it encounters an error when attempting to open a CF-netCDF dataset which contains only coordinate variables. The error message is "RuntimeError: No fields found from 1 files". This seems to be related to the CF data model's treatment of the field construct as a container for all other constructs. Having a file contain only domain information is useful in practice, particularly in the context of interpolating fields from one domain to another.

This error has been fixed in cf-python version 2.1 - thank you for pointing out this bug in the code. In this case the fixed code now does not raise an error, but simply returns no fields. New text has added at L167-L172 and L763-770 to discuss why this occurs for this interesting case, describing why cf-python (by default) finds neither fields nor a domain from this example.

L143: There is nothing in CF that says a file can't just contain coordinate variables, i.e., it's possible that N=0 and M>0. So it's not necessary that M<=N.

We disagree that a CF-compliant file can only contain coordinate variables that are associated via the dimensions of a coherent domain, because there is no explicit notion of a domain in CF, and no CF-netCDF mechanism to describe that association. The interpretation of such a file is a good example of the lack of a domain element in the CF conventions (and therefore the CF data model). This is discussed at new text L167-L172 and L763-L770.

L149, L154: t is not a coordinate variable of temp. It's a scalar coordinate variable. There is no t dimension.

The text here was indeed not clear. $t$ is a dimension of the implied domain, rather than a netCDF dimension in the file. The text has been updated to clarify this at L165-L167.

L119: CF uses the term "coordinate variable" exactly as it was defined by the NUG (as per section 1.2 in the CF convention document). The motivation for the term "auxiliary coordinate variable" was to avoid the need for the unwieldy phrase "coordinate variable in the NUG sense", i.e., auxiliary coordinate variables contain coordinate data, but are not coordinate variables.

We agree with this comment. The CF standard document is consistent in its terminology. However, it can be misunderstood by users of CF. Although incorrect, it is natural to assume that a CF auxiliary coordinate variable is a special type of CF coordinate variable. We have amended the text to clarify that this is not a mistake in the CF convention (L133-L135).

L175: I think it's misleading to describe auxiliary coordinate variables as providing "additional or alternative" coordinate information in the sense that that description makes the information sound optional. I would say that the most important use of auxiliary coordinate variables is to provide *required* coordinate information, as discussed starting in line 182.

We have amended the text where auxiliary coordinate variables are introduced to avoid the implication that they are always optional, and in the following paragraph we have made clear that they are mandatory in the case the reviewer mentions (L197-L199). That case is described after some optional uses because it needs more space to explain (L206).

Figure 8: I'm not sure what this diagram adds. L346 states it could be interpreted as a data model for CF, but I don't see how. For example there is no way from the diagram to know that a coordinate variable is a one dimensional array with a dimension name that matches the variable name and contains strictly monotonic data with no missing values. Figure 3. from Nativi et al. (2008) does a better job of expressing the CFnetCDF data model, though it too is incomplete.

Figure 8 is important as it allows our CF data model to be related diagramatically to the elements of CF which it is modelling. We think that figure 8 could be regarded as a data model of CF because it does decompose the CF into elements to which any CF-compliant dataset may be mapped, but it is not one that meets our design criteria of section 1.1. We have chosen to omit details about the nature of an element in our UML diagrams of CF-netCDF and CF so as to keep the diagrams as simple as possible. The diagrams are not meant to be stand alone, but require the supporting text in which each element is fully described. New text has been added to clarify this at L372-L373, and in the caption to figure 8.

L366: Note that the "units" attribute is not optional in most cases.

This point is included in the description of CF in section 3.8 (L336-L337), and is referenced from section 4.1 (L395).

Figure 9: Note that "Auxiliary" is misspelled in the «construct» box. Also in Figure 10.

Fixed.
* * *
**Marked-up manuscript follows:**

[revised manuscript text omitted]